# Researching COVID to enhance recovery (RECOVER) pediatric study protocol: Rationale, objectives and design

Rachel S. Gross[1]◐*, Tanayott Thaweethai[2]◐, Erika B. Rosenzweig[3], James Chan[2], Lori B. Chibnik[2], Mine S. Cicek[4], Amy J. Elliott[5], Valerie J. Flaherman[6], Andrea S. Foulkes[2], Margot Gage Witvliet[7], Richard Gallagher[8], Maria Laura Gennaro[9], Terry L. Jernigan[10,11], Elizabeth W. Karlson[12], Stuart D. Katz[13], Patricia A. Kinser[14], Lawrence C. Kleinman[15], Michelle F. Lamendola-Essel[13], Joshua D. Milner[16], Sindhu Mohandas[17], Praveen C. Mudumbi[18], Jane W. Newburger[19], Kyung E. Rhee[20], Amy L. Salisbury[14], Jessica N. Snowden[21], Cheryl R. Stein[8], Melissa S. Stockwell[22,23], Kelan G. Tantisira[24], Moriah E. Thomason[8], Dongngan T. Truong[25], David Warburton[26], John C. Wood[27], Shifa Ahmed[2], Almary Akerlundh[28], Akram N. Alshawabkeh[29], Brett R. Anderson[3], Judy L. Aschner[30], Andrew M. Atz[31], Robin L. Aupperle[32], Fiona C. Baker[33], Venkataraman Balaraman[34], Dithi Banerjee[35], Deanna M. Barch[36], Arielle Baskin-Sommers[37], Sultana Bhuiyan[13], Marie-Abele C. Bind[2], Amanda L. Bogie[38], Tamara Bradford[39], Natalie C. Buchbinder[40], Elliott Bueler[13], Hülya Bükülmez[41], B. J. Casey[42], Linda Chang[43], Maryanne Chrisant[44], Duncan B. Clark[45], Rebecca G. Clifton[46], Katharine N. Clouser[30], Lesley Cottrell[47], Kelly Cowan[48], Viren D'Sa[49], Mirella Dapretto[50], Soham Dasgupta[51], Walter Dehority[52], Audrey Dionne[19], Kirsten B. Dummer[53], Matthew D. Elias[54], Shari Esquenazi-Karonika[13], Danielle N. Evans[55], E. Vincent S. Faustino[56], Alexander G. Fiks[57], Daniel Forsha[58], John J. Foxe[59], Naomi P. Friedman[60], Greta Fry[61], Sunanda Gaur[15], Dylan G. Gee[37], Kevin M. Gray[62], Stephanie Handler[63], Ashraf S. Harahsheh[64], Keren Hasbani[65], Andrew C. Heath[66], Camden Hebson[67], Mary M. Heitzeg[68], Christina M. Hester[69], Sophia Hill[13], Laura Hobart-Porter[70], Travis K. F. Hong[34], Carol R. Horowitz[71], Daniel S. Hsia[72], Matthew Huentelman[73], Kathy D. Hummel[74], Katherine Irby[74], Joanna Jacobus[75], Vanessa L. Jacoby[76], Pei-Ni Jone[77], David C. Kaelber[78,79], Tyler J. Kasmarcak[80], Matthew J. Kluko[56], Jessica S. Kosut[34], Angela R. Laird[81], Jeremy Landeo-Gutierrez[82], Sean M. Lang[83], Christine L. Larson[84], Peter Paul C. Lim[85], Krista M. Lisdahl[84], Brian W. McCrindle[86], Russell J. McCulloh[87], Kimberly McHugh[80], Alan L. Mendelsohn[88], Torri D. Metz[89], Julie Miller[90], Elizabeth C. Mitchell[91], Lerraughn M. Morgan[92], Eva M. Müller-Oehring[33], Erica R. Nahin[13], Michael C. Neale[93], Manette Ness-Cochinwala[15], Sheila M. Nolan[94], Carlos R. Oliveira[56], Onyekachukwu Osakwe[95], Matthew E. Oster[96], R. Mark Payne[97], Michael A. Portman[98], Hengameh Raissy[99], Isabelle G. Randall[13], Suchitra Rao[100], Harrison T. Reeder[2], Johana M. Rosas[13], Mark W. Russell[101], Arash A. Sabati[102], Yamuna Sanil[103], Alice I. Sato[104], Michael S. Schechter[105], Rangaraj Selvarangan[35], S. Kristen Sexson Tejtel[106], Divya Shakti[95], Kavita Sharma[107], Lindsay M. Squeglia[62], Shubika Srivastava[108], Michelle D. Stevenson[109], Jacqueline Szmuszkovicz[110], Maria M. Talavera-Barber[111], Ronald J. Teufel, II[31], Deepika Thacker[108], Felicia Trachtenberg[90], Mmekom M. Udosen[18], Megan R. Warner[28], Sara E. Watson[109], Alan Werzberger[112], Jordan C. Weyer[113], Marion J. Wood[18], H. Shonna Yin[114], William T. Zempsky[115], Emily Zimmerman[116], Benard P. Dreyer[1], on behalf of the RECOVER-Pediatric Consortium¶

1 Department of Pediatrics, New York University Grossman School of Medicine, New York, New York, United States of America, 2 Department of Biostatistics, Massachusetts General Hospital, Boston, Massachusetts, United States of America, 3 Division of Pediatric Cardiology, Department of Pediatrics, Columbia University Vagelos College of Physicians and Surgeons and NewYork-Presbyterian, New York, New York, United States of America, 4 Department of Laboratory Medicine and Pathology, Mayo Clinic Hospital, Rochester, Minnesota, United States of America, 5 Avera Research Institute, Avera Health, Sioux Falls, South Dakota, United States of America, 6 Department of Pediatrics, University of California San Francisco, San Francisco,

**Data Availability Statement:** No datasets were generated or analysed during the current study. All

relevant data from this study will be made available upon study completion.

**Funding:** National Institutes of Health (NIH) Agreement OTA OT2HL161847 (SDK, RG), OT2HL161841 (ASF). https://www.nih.gov/ The funders did not and will not have a role in study design, data collection and analysis, decision to publish, or preparation of the manuscript.

**Competing interests:** I have read the journal's policy and the authors of this manuscript have the following competing interests: Brett Anderson reported receiving direct support for work not related to RECOVER work/publications from Genentech and the National Institute of Allergy and Immunology. Walter Dehority reported receiving grant support from Merck and participating in research for the Moderna COVID-19 pediatric vaccine trial and the Pfizer Paxlovid trial. Alex Fiks reported receiving support from NJM insurance and personal consulting fees not related to this paper from Rutgers University and the American Academy of Pediatrics. Ashraf Harahsheh reported serving as a scientific advisory board member unrelated to this paper for OP2 DRUGS. Lawrence Kleinman reported serving as an unpaid member of the Board of Directors for the DARTNet Institute, as a principle investigator at Quality Matters, Inc., and as the Vice Chair for the Borough of Metuchen Board of Health. Dr. Kleinman also reported grant support for work not related to RECOVER work/ publications from NIH, HRSA, and the Robert Wood Johnson Foundation. Dr. Kleinman also reported minority individual stock ownership in Apple Computer, Sanofi SA, Experion, GlaxoSmithKline, Magyar Bank, Regeneron Pharmaceuticals, JP Morgan Chase, and Amgen Inc. Torri Metz reported participating as a Principle Investigator in the medical advisory board for the planning of a Pfizer clinical trial of SARS-CoV-2 vaccination in pregnancy. She is also a principle investigator for a Pfizer study evaluating the pharmacokinetics of Paxlovid in pregnant people with COVID-19. Joshua Milner reported serving as a member of the Scientific Advisory Board for Blueprint Medicines, in a capacity unrelated to RECOVER work/publications. This does not alter our adherence to PLOS ONE policies on sharing data and materials.

California, United States of America, **7** Department of Sociology, Lamar University, Beaumont, Texas, United States of America, **8** Department of Child and Adolescent Psychiatry, New York University Grossman School of Medicine, New York, New York, United States of America, **9** Public Health Research Institute and Department of Medicine, Rutgers New Jersey Medical School, Newark, New Jersey, United States of America, **10** Center for Human Development, Department of Cognitive Science, University of California San Diego, San Diego, California, United States of America, **11** Departments of Psychiatry and Radiology, University of California San Diego School of Medicine, San Diego, California, United States of America, **12** Department of Medicine, Harvard Medical School, Boston, Massachusetts, United States of America, **13** Department of Medicine, New York University Grossman School of Medicine, New York, New York, United States of America, **14** School of Nursing, Virginia Commonwealth University, Richmond, Virginia, United States of America, **15** Department of Pediatrics, Rutgers Robert Wood Johnson Medical School, New Brunswick, New Jersey, United States of America, **16** Division of Pediatric Allergy, Department of Pediatrics, Immunology and Rheumatology, Columbia University Vagelos College of Physicians and Surgeons and NewYork-Presbyterian, New York, New York, United States of America, **17** Department of Infectious Diseases, Children's Hospital Los Angeles, Keck School of Medicine, University of Southern California, Los Angeles, California, United States of America, **18** Department of Population Health, New York University Grossman School of Medicine, New York, New York, United States of America, **19** Department of Cardiology, Boston Children's Hospital, Boston, Massachusetts, United States of America, **20** Division of Child and Community Health, Department of Pediatrics, University of California San Diego School of Medicine, San Diego, California, United States of America, **21** Departments of Pediatrics and Biostatistics, University of Arkansas for Medical Sciences, Little Rock, Arkansas, United States of America, **22** Division of Child and Adolescent Health, Department of Pediatrics, Columbia University Vagelos College of Physicians and Surgeons and NewYork-Presbyterian, New York, New York, United States of America, **23** Department of Population and Family Health, Columbia University Mailman School of Public Health, New York, New York, United States of America, **24** Division of Pediatric Respiratory Medicine, Department of Pediatrics, University of California San Diego School of Medicine, San Diego, California, United States of America, **25** Division of Pediatric Cardiology, University of Utah and Primary Children's Hospital, Salt Lake City, Utah, United States of America, **26** Division of Neonatology, Department of Pediatrics, Children's Hospital Los Angeles, Keck School of Medicine, University of Southern California, Los Angeles, California, United States of America, **27** Department of Pediatrics and Radiology, Children's Hospital Los Angeles, Los Angeles, California, United States of America, **28** Department of Pulmonary Research, Rady Children's Hospital-San Diego, San Diego, California, United States of America, **29** College of Engineering, Northeastern University, Boston, Massachusetts, United States of America, **30** Department of Pediatrics, Hackensack Meridian School of Medicine, Nutley, New Jersey, United States of America, **31** Department of Pediatrics, Medical University of South Carolina, Charleston, South Carolina, United States of America, **32** Oxley College of Health Sciences, Laureate Institute for Brain Research, Tulsa, Oklahoma, United States of America, **33** Department of Biosciences, SRI International, Menlo Park, California, United States of America, **34** Department of Pediatrics, Kapiolani Medical Center for Women and Children, Honolulu, Hawaii, United States of America, **35** Department of Pathology and Laboratory Medicine, Children's Mercy Hospital, Kansas City, Missouri, United States of America, **36** Department of Psychiatry, Washington University in St. Louis, Saint Louis, Missouri, United States of America, **37** Department of Psychology, Yale University School of Medicine, New Haven, Connecticut, United States of America, **38** Department of Pediatrics, University of Oklahoma Health Science Center, Oklahoma City, Oklahoma, United States of America, **39** Division of Pediatric Cardiology, Department of Pediatrics, Children's Hospital of New Orleans and LSU Health Sciences Center, New Orleans, United States of America, **40** Center for Human Development, University of California San Diego, San Diego, California, United States of America, **41** Division of Rheumatology, Department of Pediatrics, The MetroHealth System, Case Western Reserve University, Cleveland, Ohio, United States of America, **42** Department of Neuroscience and Behavior, Barnard College—Columbia University, New York, New York, United States of America, **43** Departments of Diagnostic Radiology & Nuclear Medicine and Neurology, University of Maryland Baltimore, Baltimore, Maryland, United States of America, **44** Department of Women's and Children's Health, Charles E. Schmidt College of Medicine at Florida Atlantic University, Hollywood, Florida, United States of America, **45** Department of Psychiatry, University of Pittsburgh Medical Center, Pittsburgh, Pennsylvania, United States of America, **46** Biostatistics Center, George Washington University, Washington, DC, United States of America, **47** Department of Pediatrics, West Virginia University, Morgantown, West Virginia, United States of America, **48** Department of Pediatrics, Robert Larner M.D. College of Medicine at the University of Vermont, Burlington, Vermont, United States of America, **49** Department of Pediatrics, Rhode Island Hospital, Providence, Rhode Island, United States of America, **50** Department of Psychiatry and Biobehavioral Sciences, University of California Los Angeles, Los Angeles, California, United States of America, **51** Department of Pediatrics, Norton Children's Hospital, University of Louisville, Louisville, Kentucky, United States of America, **52** Division of Infectious Diseases, Department of Pediatrics, University of New Mexico, Albuquerque, New Mexico, United States of America, **53** Department of Pediatrics, University of California San Diego, San Diego, California, United States of America, **54** Division of Cardiology, Children's Hospital of Philadelphia, Philadelphia, Pennsylvania, United States of America,

**55** Arkansas Children's Research Institute, Arkansas Children's Hospital, Little Rock, Arkansas, United States of America, **56** Department of Pediatrics, Yale University School of Medicine, New Haven, Connecticut, United States of America, **57** Department of Pediatrics, Children's Hospital of Philadelphia, Philadelphia, Pennsylvania, United States of America, **58** Department of Cardiology, Children's Mercy Kansas City, Ward Family Heart Center, Kansas City, Missouri, United States of America, **59** Department of Neuroscience, University of Rochester School of Medicine and Dentistry, Rochester, New York, United States of America, **60** Institute for Behavioral Genetics and Department of Psychology and Neuroscience, University of Colorado Boulder, Bolder, Colorado, United States of America, **61** Pennington Biomedical Research Center Clinic, Pennington Biomedical Research Center, Baton Rouge, Louisiana, United States of America, **62** Department of Psychiatry and Behavioral Sciences, Medical University of South Carolina, Charleston, South Carolina, United States of America, **63** Division of Pediatric Cardiology, Department of Pediatrics, Medical College of Wisconsin, Milwaukee, Wisconsin, United States of America, **64** Division of Cardiology, Department of Pediatrics, George Washington University School of Medicine & Health Sciences, Washington, DC, United States of America, **65** Division of Pediatric Cardiology, Department of Pediatrics, Dell Children's Medical Center, Dell Medical School, Austin, Texas, United States of America, **66** Department of Psychiatry, Washington University School of Medicine, St Louis, Missouri, United States of America, **67** Division of Pediatric Cardiology, Department of Pediatrics, University of Alabama at Birmingham, Birmingham, Alabama, United States of America, **68** Department of Psychiatry, University of Michigan, Ann Arbor, Michigan, United States of America, **69** Division of Practice-Based Research, Innovation, & Evaluation, American Academy of Family Physicians, Leawood, Kansas, United States of America, **70** Departments of Pediatrics and Physical Medicine & Rehabilitation, Section of Pediatric Rehabilitation, University of Arkansas for Medical Sciences, Little Rock, Arkansas, United States of America, **71** Center for Health Equity and Community Engaged Research and Department of Population Health Science and Policy, Icahn School of Medicine at Mount Sinai, New York, New York, United States of America, **72** Clinical Trials Unit, Pennington Biomedical Research Center, Baton Rouge, Louisiana, United States of America, **73** Division of Neurogenomics, Translational Genomics Research Institute, Phoenix, Arizona, United States of America, **74** Department of Pediatrics, Arkansas Children's Hospital, University of Arkansas Medical School, Little Rock, Arkansas, United States of America, **75** Department of Psychiatry, University of California San Diego, San Diego, California, United States of America, **76** Department of Obstetrics, Gynecology, and Reproductive Sciences, University of California San Francisco, San Francisco, California, United States of America, **77** Department of Pediatrics, Pediatric Cardiology, Lurie Children's Hospital, Northwestern University Feinberg School of Medicine, Chicago, Illinois, United States of America, **78** The Center for Clinical Informatics Research and Education, The MetroHealth System and the Departments of Pediatrics, Internal Medicine, and Population and Quantitative Health Sciences, Case Western Reserve University, Cleveland, Ohio, United States of America, **79** Departments of Pediatrics, Internal Medicine, and Population & Quantitative Health Sciences, Case Western Reserve University, Cleveland, Ohio, United States of America, **80** Department of Pediatric Clinical Research, Medical University of South Carolina, Charleston, South Carolina, United States of America, **81** Department of Physics, Florida International University, Miami, Florida, United States of America, **82** Respiratory Medicine Division, Department of Pediatrics, University of California San Diego, San Diego, California, United States of America, **83** Heart Institute, Cincinnati Children's Hospital Medical Center, Cincinnati, Ohio, United States of America, **84** Department of Psychology, University of Wisconsin-Milwaukee, Milwaukee, Wisconsin, United States of America, **85** Department of Pediatric Infectious Disease, Avera McKennan University Health Center, University of South Dakota, Sioux Falls, South Dakota, United States of America, **86** Department of Pediatrics, University of Toronto, Labatt Family Heart Center, The Hospital for Sick Children, Toronto, Ontario, Canada, **87** Department of Pediatrics, University of Nebraska Medical Center, Omaha, Nebraska, United States of America, **88** Division of Developmental-Behavioral Pediatrics, Department of Pediatrics, New York University Grossman School of Medicine, New York, New York, United States of America, **89** Department of Obstetrics and Gynecology, University of Utah Health, Salt Lake City, Utah, United States of America, **90** Carelon Research, Newton, Massachusetts, United States of America, **91** Division of Pediatric Cardiology, Department of Pediatrics, Cohen Children's Medical Center (Northwell Health), New Hyde Park, New York, United States of America, **92** Department of Pediatrics, Valley Children's Healthcare, Madera, California, United States of America, **93** Virginia Institute for Psychiatric and Behavioral Genetics, Virginia Commonwealth University, Richmond, Virginia, United States of America, **94** Department of Pediatrics, New York Medical College, Valhalla, New York, United States of America, **95** Division of Pediatric Cardiology, Department of Pediatrics, University of Mississippi Medical Center, Jackson, Mississippi, United States of America, **96** Department of Pediatric Cardiology, Children's Healthcare of Atlanta, Atlanta, Georgia, United States of America, **97** Division of Pediatric Cardiology, Department of Pediatrics, Riley Hospital for Children, Indiana University School of Medicine, Indianapolis, Indiana, United States of America, **98** Division of Cardiology, Department of Pediatrics, Seattle Children's and University of Washington, Seattle, Washington, United States of America, **99** Department of Pediatrics, University of New Mexico, Health Sciences Center, Albuquerque, New Mexico, United States of America, **100** Division of Infectious Diseases, Department of Pediatrics, Epidemiology and Hospital Medicine, University of Colorado Anschutz Medical Campus, Aurora, Colorado, United States of America,

**101** Department of Pediatrics, University of Michigan Health System, Ann Arbor, Michigan, United States of America, **102** Department of Pediatric Cardiology, Phoenix Children's Hospital, Phoenix, Arizona, United States of America, **103** Division of Pediatric Cardiology, Department of Pediatrics, Children's Hospital of Michigan, Detroit, Michigan, United States of America, **104** Department of Pediatric Infectious Disease, University of Nebraska Medical Center, Omaha, Nebraska, United States of America, **105** Department of Pediatrics, Children's Hospital of Richmond at Virginia Commonwealth University, Richmond, Virginia, United States of America, **106** Division of Pediatric Cardiology, Department of Pediatrics, Texas Children's Hospital, Baylor College of Medicine, Houston, Texas, United States of America, **107** Department of Pediatrics, University of Texas Southwestern Medical Center, Dallas, Texas, United States of America, **108** Division of Cardiovascular Medicine, Department of Pediatric Cardiology, Nemours Children's Health, Wilmington, Delaware, United States of America, **109** Department of Pediatrics, University of Louisville School of Medicine, Louisville, Kentucky, United States of America, **110** Division of Cardiology, Children's Hospital Los Angeles, Los Angeles, California, United States of America, **111** Department of Pediatrics, Avera McKennan Hospital and University Health Center, Sioux Falls, South Dakota, United States of America, **112** Department of Pediatrics, Columbia University Vagelos College of Physicians and Surgeons, New York, New York, United States of America, **113** Center for Individualized Medicine, Mayo Clinic Hospital, Rochester, Minnesota, United States of America, **114** Departments of Pediatrics and Population Health, New York University Grossman School of Medicine, New York, New York, United States of America, **115** Department of Pediatrics, Connecticut Children's Medical Center, University of Connecticut School of Medicine, Hartford, Connecticut, United States of America, **116** Department of Communication Sciences & Disorders, Northeastern University, Boston, Massachusetts, United States of America

☯ These authors contributed equally to this work.
¶ Membership of the RECOVER-Pediatrics Consortium is provided in the Supplement S1 Appendix.
* Rachel.Gross@nyulangone.org

# Abstract

## Importance

The prevalence, pathophysiology, and long-term outcomes of COVID-19 (post-acute sequelae of SARS-CoV-2 [PASC] or "Long COVID") in children and young adults remain unknown. Studies must address the urgent need to define PASC, its mechanisms, and potential treatment targets in children and young adults.

## Observations

We describe the protocol for the Pediatric Observational Cohort Study of the NIH's **RE**searching **COV**ID to **E**nhance **R**ecovery (RECOVER) Initiative. RECOVER-Pediatrics is an observational meta-cohort study of caregiver-child pairs (birth through 17 years) and young adults (18 through 25 years), recruited from more than 100 sites across the US. This report focuses on two of four cohorts that comprise RECOVER-Pediatrics: 1) a *de novo* RECOVER prospective cohort of children and young adults with and without previous or current infection; and 2) an extant cohort derived from the Adolescent Brain Cognitive Development (ABCD) study ($n$ = 10,000). The *de novo* cohort incorporates three tiers of data collection: 1) remote baseline assessments (Tier 1, n = 6000); 2) longitudinal follow-up for up to 4 years (Tier 2, n = 6000); and 3) a subset of participants, primarily the most severely affected by PASC, who will undergo deep phenotyping to explore PASC pathophysiology (Tier 3, n = 600). Youth enrolled in the ABCD study participate in Tier 1. The pediatric protocol was developed as a collaborative partnership of investigators, patients, researchers, clinicians, community partners, and federal partners, intentionally promoting inclusivity and diversity. The protocol is adaptive to facilitate responses to emerging science.

## Conclusions and relevance

RECOVER-Pediatrics seeks to characterize the clinical course, underlying mechanisms, and long-term effects of PASC from birth through 25 years old. RECOVER-Pediatrics is designed to elucidate the epidemiology, four-year clinical course, and sociodemographic correlates of pediatric PASC. The data and biosamples will allow examination of mechanistic hypotheses and biomarkers, thus providing insights into potential therapeutic interventions.

## Clinical trials.gov identifier

Clinical Trial Registration: http://www.clinicaltrials.gov. Unique identifier: NCT05172011.

## Introduction

Long COVID, or the post-acute sequelae of SARS-CoV-2 (PASC), has been defined as symptoms, signs and conditions that continue or develop after a SARS-CoV-2 infection. These symptoms can affect people for weeks, months or even years after getting coronavirus disease 2019 (COVID-19) [1, 2]. Symptoms can develop shortly after the initial recovery from an acute COVID-19 episode or persist from the initial illness. Symptoms may also emerge later or fluctuate or relapse over time. These symptoms can have debilitating effects on the daily health and quality of life of those affected.

The COVID-19 pandemic has significantly impacted child health. Nearly 100 million people have been diagnosed with COVID-19 in the United States (US), with nearly 16 million children [3]. Although it is estimated that between 10% and 30% of adults experience persistent symptoms from COVID-19 [4], the prevalence in children is less well-established [5, 6]. As an emerging illness, the absence of universally-accepted PASC definitions in children challenge the elucidation of its epidemiology.

Unique challenges in understanding PASC symptoms in children have likely contributed to the limited evidence. For example, young children might not be able to articulate their symptoms. This has required studies to rely on caregiver interpretation of their young child's symptoms. In addition, manifestation of symptoms may vary substantively across stages of physiological, emotional, and cognitive development [7]. As the medical community shifts from managing serious acute disease to addressing long-term consequences, large scale studies are needed to define PASC in children across the life course, to understand its natural history, and to develop evidence to guide successful treatment.

The pandemic began with a misconception that children were spared [8]. We now recognize that children and families are greatly impacted during both acute and chronic phases [9–12]. One distinct manifestation in children was recognized in April 2020; now called Multisystem Inflammatory Syndrome in Children (MIS-C) [13]. This debilitating hyperinflammatory syndrome has impacted over 9,000 children and young adults in the US [14], and represents a distinct post-acute syndrome that is typically recognizable in clinical practice. Other more chronic manifestations of PASC are challenging to characterize and identify. Furthermore, children with PASC may present with different symptoms and greater mental health concerns than adults [3, 15–21]. Additional phenotypes of childhood PASC are being reported, including phenotypes similar to postural orthostatic tachycardia syndrome (POTS), myalgic encephalomyelitis/chronic fatigue syndrome (ME/CFS), postintensive care unit syndrome, and potentially many others [22–24]. Therefore, a compelling rationale exists to invest resources and effort to study PASC in children. The NIH's REsearching COVID to Enhance Recovery

(RECOVER) Initiative responded by bringing together researchers, communities, and families in a systematic study of PASC in children [25]. Evidence that leads to improved health trajectories of children with PASC, could have population-level health impacts for decades to come.

## Study rationale

RECOVER has established the Pediatric Observational Cohort Study (RECOVER-Pediatrics), which is a combined retrospective and prospective longitudinal study, including four distinct cohorts, integrated together as a meta-cohort [25]. The overall goal is to characterize the clinical course, underlying mechanisms and long-term health effects of PASC on children and young adults from birth through 25 years old, to inform future pediatric preventive and treatment measures.

## Study aims

RECOVER-Pediatrics scientific aims are to:

1. Characterize the prevalence and incidence of new onset or worsening symptoms related to PASC

2. Characterize the spectrum of clinical symptoms of PASC, including distinct phenotypes, and describe the clinical course and recovery.

3. Identify risk and resiliency factors for developing PASC and recovering from PASC.

4. Define the pathophysiology of PASC, including subclinical organ dysfunction, and identify biological mechanisms underlying the pathogenesis of PASC.

# Materials and methods

## Overview of study design

RECOVER-Pediatrics is a longitudinal, observational meta-cohort study of children and young adults (ages birth through 25 years) and their caregivers, recruited from healthcare- and community-based settings in more than 100 sites throughout the US, including Puerto Rico. Those *with and without* a history of a SAR-CoV-2 infection are included. For those 17 years or younger, data are collected by caregiver report and child direct assessments, and for those 18 through 25 years old by self-report. The study is being conducted from March 2022 to March 2026.

The pediatric meta-cohort is comprised of four distinct cohorts: 1) *de novo RECOVER prospective cohort* including children and young adults ages birth through 25 years, with or without a known history of infection, and their caregivers; 2) *Adolescent Brain Cognitive Development (ABCD)* extant cohort, the largest long-term US study of brain development in adolescence [26, 27]; 3) *In utero exposure cohort*, including children less than 3 years old born to individuals with and without a SAR-CoV-2 infection during pregnancy [28, 29]; and 4) *COVID MUSIC Study* extant cohort (*Long-Term Outcomes after the Multisystem Inflammatory Syndrome In Children*), including children and young adults with history of MIS-C [30]. This report focuses on the *de novo* cohort and ABCD (Fig 1).

Fig 1 shows a tiered overview of 2 of the 4 cohorts included in the meta-cohort (*de novo* RECOVER prospective cohort and ABCD), their participation in the three study tiers, and their targeted sample sizes (see *Study Participants*). Children and young adults ages newborn through 25 years old will be enrolled in the meta-cohort at Tier 1 for the *de novo* RECOVER

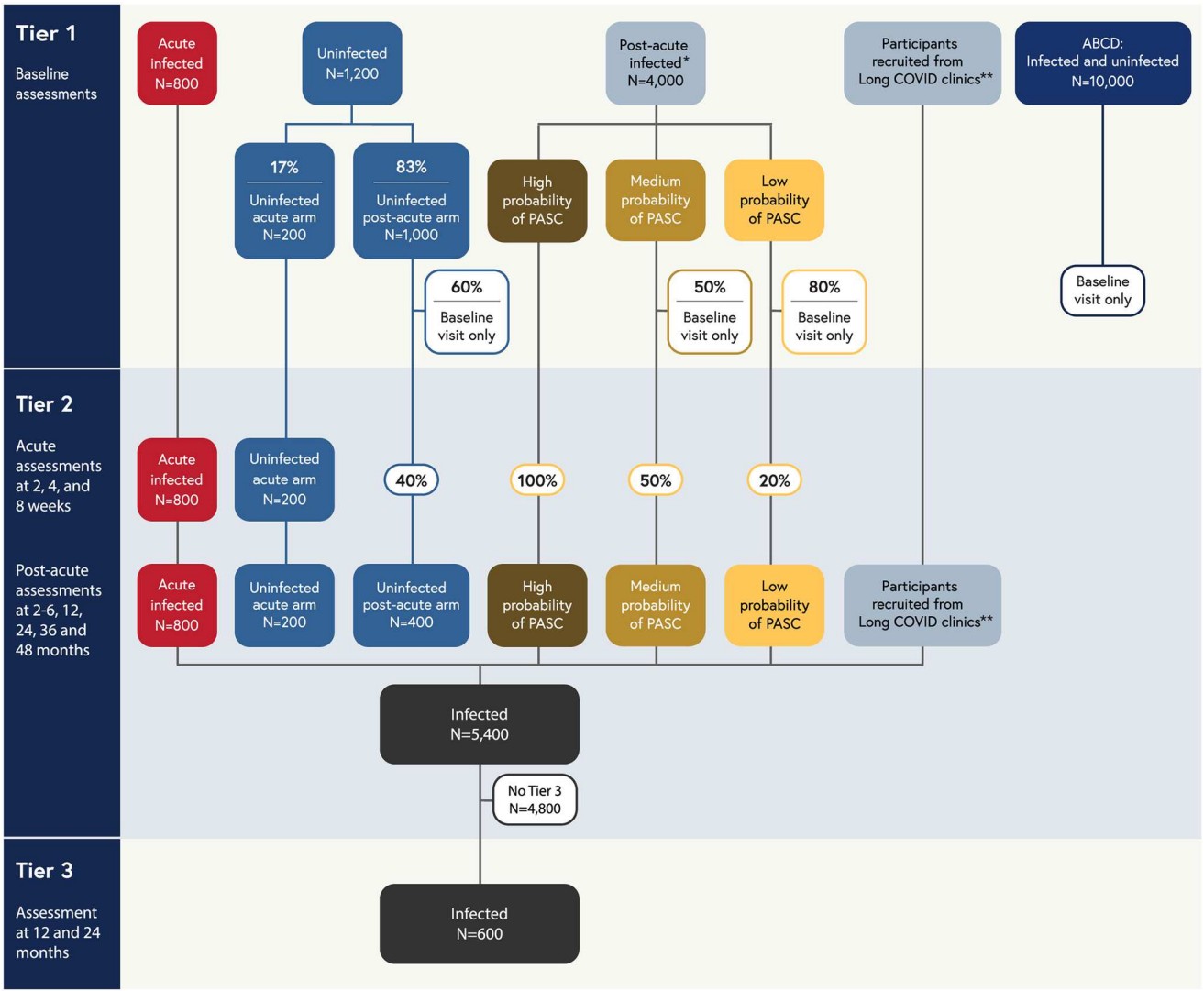

**Fig 1. Overview of RECOVER-Pediatrics (de novo and ABCD cohorts).**

prospective cohort (more than 6,000 from birth through 25 years old, including those with and without history of infection), and from ABCD (up to 10,000 adolescents with and without history of infection). All children and young adults enrolled in the study complete a baseline assessment (Tier 1). Percentages shown indicate random sampling proportions. Children and young adults without history of infection are assigned at random with prespecified proportions to the acute and post-acute arms. All children and young adults with history of infection who enroll into the acute arm and those without a history of infection who are randomized to the acute arm are asked to complete assessments at 2, 4, and 8 weeks. Following a promotion algorithm, children and young adults in Tier 1 will be selected to be promoted to Tier 2, which includes assessments at 2–6, 12, 24, 36, and 48 months after enrollment. 600 children and young adults with history of infection, selected from Tier 2, will complete more intensive Tier 3 assessments at 12 and 24 months after enrollment.

*Children and young adults with history of infection who enroll in the post-acute arm ("post-acute infected", n = 4,000) are stratified into High, Medium, and Low probability of

PASC groups based on a combination of past Long COVID diagnoses, Tier 1 Global PROMIS health measures, and symptom survey screener responses. Then, 100% of the high probability group, 50% of the medium probability group, and 20% of the low probability group are promoted at random to Tier 2. In June 2023, the promotion rate for the medium probability group was increased to 100% to enhance promotion rates and include the wider spectrum of symptom severity within our longitudinal cohort. Since the distribution of these probability groups is unknown a priori, sample sizes are not specified for each category. Overall, the number of children and young adults who progress to Tier 2 will be less than the initial post-acute infected sample size, but the total target sample size for infected children and young adults in Tier 2 is 5,400.

** In order to achieve a sample of 5,400 children and young adults with history of infection in Tier 2 that is skewed towards those with greater likelihood of having PASC, additional children and young adults will be recruited from Long COVID clinics and subspecialty services to complete both Tier 1 and Tier 2 assessments.

RECOVER-Pediatrics is structured in a sequential fashion with three Tiers of data collection. Participants are enrolled initially into Tier 1, which consists of a broad screening of health using remote surveys and biospecimen collection. Participants may subsequently progress to Tier 2, which includes a detailed review of health collected longitudinally for up to four years, using a combination of remote surveys and in-person assessments of biological and psychosocial data. In order to achieve the sample size required for Tier 2 assessments, other study participants will be recruited who present with a high probability of having PASC, such as those directly recruited from a clinic that focuses on Long COVID or presenting with a physician diagnosis of PASC. These participants will receive Tier 1 assessments and progress directly to Tier 2. Finally, in Tier 3, a subset of children and young adults most severely affected by PASC will undergo deep phenotyping with more intensive assessments to study PASC pathophysiology.

RECOVER-Pediatrics Tier 1 assessments aim to characterize the prevalence and incidence of new onset or worsening of sustained COVID-related symptoms (aim 1) and to gain a comprehensive understanding of the impact of exposure to a SARS-CoV-2 infection on broad physical, behavioral and mental health (aim 2). Tier 2 facilitates studying the natural history of PASC symptoms and potential recovery over time (aim 2). Child, household, and caregiver factors gathered in Tier 1, such as social determinants of health and prior health conditions, will be assessed to determine how they increase the risk of or protect against specific clinical outcomes (aim 3). Finally, Tier 3 data investigates long-term effects on multiple organ systems and child development (aim 4). Additionally, integration of Tier 1 and Tier 2 data will allow investigation of COVID-disease exposures and experiences which may be responsible for the clinical patterns observed in Tier 3.

The pediatric protocol was designed through collaboration across key stakeholders, including patients, caregivers, researchers, clinicians, community partners, and federal partners, fostering a patient-centered approach and promoting inclusivity and diversity. The pediatric protocol is adaptive to facilitate the changes needed in light of emerging science and the evolving pandemic.

## Study organizational structure and management

Study infrastructure includes four cores: 1) Clinical Science Core (CSC) at the NYU Grossman School of Medicine, which oversees study sites and provides scientific leadership in collaboration with hub and site Principal Investigators; 2) Data Resource Core (DRC) at Massachusetts General Hospital and Brigham and Women's Hospital, which provides scientific and statistical

leadership, and handles data management and storage; 3) PASC Biorepository Core (PBC) at Mayo Clinic, which manages biospecimens obtained; and 4) Administrative Coordinating Center (ACC) at RTI International, which provides operational and administrative support; collectively these form the Core Operations Group. The four cores are supported by oversight committees and pathobiology task forces provide content-specific input. RECOVER cohort studies are overseen by the National Community Engagement Group (NCEG) composed of patient and community representatives, a Steering Committee composed of site Principal Investigators and NIH program leadership, an Executive Committee composed of NIH Institute leadership, and an Observational Safety Monitoring Board composed of experts in longitudinal observational studies, epidemiology, bioethics, and biostatistics. RECOVER-Pediatrics includes 10 hubs that manage ~100 sites (S1 Table), located in more than 39 states, Washington DC and Puerto Rico. Awardees were selected through a process that included independent peer review in response to OTA-21-015B.

## Ethics

The study was approved by the NYU Grossman School of Medicine Institutional Review Board (IRB), which serves as the single IRB for the majority of the study sites. A few pre-existing networks use their own central IRBs through an exemption granted by the NIH (e.g., ABCD, MUSIC). Caregivers, for children 17 years old or younger, and young adult participants provide signed informed consent to participate.

## Recruitment, consent, and screening strategies

The *de novo RECOVER prospective cohort study* is recruiting participants from healthcare- and community-based settings. Healthcare-based recruitment involves local media, text messaging, hospital websites, COVID registries, and partnerships with pediatric practices, nurse hotlines, or emergency departments. Community-based recruitment includes partnering with community health workers, school nurses, sports coaches, health fairs, and a mobile van to access rural communities. Participants can also join by self-referral through the RECOVER website, or in response to plain language and picture-based recruitment materials in both English and Spanish, which were developed with community input and using health literacy principles [31].

Eligible dyads complete an informed written consent process at enrollment for Tiers 1 and 2. The consent process may be conducted using telephone, a secure video conference platform approved for exchange of PHI, or in person (using either a signed written consent form or via electronic informed consent [e-consenting]). An assent process is being conducted for children between 7 years and 17 years old. The study team explains the assent document to the child and parent/legal guardian and answers all questions. Child understanding of the key elements of the assent document is assessed by the study team and parent/legal guardian. The child either signs the age-appropriate assent document or provides verbal assent (with documentation in the local records and the central REDcap). Young adults, aged 18 through 25 years old, sign their own informed consent. Tier 3 consent forms will only be completed when testing is offered. A standardized teach back method is implemented as needed to ensure understanding of the key aspects of participation before enrollment. Participants are reconsented if there are major changes to the study design or to anticipated risks.

In ABCD, 11,880 children aged 9–10 years old were recruited from community and school sites to participate in a 10-year study with the goal of understanding neurocognitive development during adolescence [26, 27, 32]. All ABCD participants are being contacted and offered enrollment into RECOVER-Pediatrics.

## Eligibility criteria

Children and young adults from birth through 25 years old are eligible to be enrolled in the *de novo cohort*, regardless of history of SARS CoV-2 infection. Enrolled participants are then categorized as either "infected" or "uninfected": Infected participants have history of suspected, probable, or confirmed SARS-CoV-2 infection, defined by the World Health Organization (WHO) criteria [33], evidence of infection by serum antibody profile, or a history of MIS-C. Uninfected participants are those who self-report as having no history of a SARS-CoV-2 infection and who have never met WHO criteria; they have no evidence of a past asymptomatic infection in their medical history or evidence of past infection by serum antibody profile.

A primary caregiver, defined as an individual responsible for the enrolled child or young adult who resides in the same household, such as biological or nonbiological family member, is invited to enroll.

The primary exclusion criterion is any child or young adult with co-morbid illness with expected survival of less than 2 years. There is no limit to the number of children or young adults who can be enrolled from a single household. See supplemental tables for detailed eligibility criteria, definitions of analytic groups, and the World Health Organization Criteria (S2–S4 Tables).

## Study participants

Recruitment is striving for a diverse sample that generally represents the US population, and encourages participation from rural or medically underserved communities, non-English speaking participants, and non-hospitalized participants with an acute COVID-19 infection. Participants are compensated for completing assessments and reimbursed for excess travel.

At least 6,000 participants will be recruited into the *de novo* cohort (Fig 1). Children and young adults with history of infection are classified into *one of two study arms* (acute arm vs. post-acute arm), based on their history of SARS-CoV-2 infection and infection dates. The *acute arm* includes 800 children and young adults whose most recent SARS-CoV-2 infection was 30 days or less prior to enrollment. The *post-acute arm* includes 4,000 children and young adults whose most recent SARS-CoV-2 infection was greater than 30 days prior to enrollment. In the group without a history of infection, 1,200 children and young adults will be randomly assigned to follow either the acute (200, or 17%) or post-acute (1,000 or 83%) arm of the protocol. Additional children and young adults will be recruited from Long COVID clinics and other subspecialty services in order to achieve Tier 2 sample size targets (see *Timing of Study Assessments*).

Up to 10,000 participants will also be recruited from the ABCD cohort.

## Timing of study assessments

The assessments for the *de novo cohort* consists of three tiers, which vary in timing, collection methods and intensity.

*Tier 1* (baseline visit for all participants) includes a single visit that is completed either via self-administration (remote and electronic) or research staff-assisted collection (e.g., telephone, videoconference, or in-person).

*Tier 2* (follow-up visits) includes five longitudinal in-person visits at 2 to 6-, 12-, 24-, 36- and 48-months post-enrollment. The children and young adults followed longitudinally in Tier 2 are selected based on a sampling scheme that prioritizes the acute arm as well as children and youth in the post-acute arm with a greater likelihood of having PASC. Promotion to Tier 2 occurs as follows: 1) All children/young adults in the *acute arm* with or without history of infection will be promoted; 2) children/young adults in the *post-acute arm with a history of*

*infection* will be promoted at a rate dependent on their likelihood of PASC based on prior Long COVID diagnoses, Tier 1 PROMIS global health measure responses [34–36], and symptoms screener survey responses [18, 37] (Table 1); and 3) 40% of children/young adults without known infection in the post-acute arm, selected at random, will be promoted. In addition to promoting children and young adults from Tier 1, children and young adults will also be recruited from Long COVID clinics and subspecialty services to achieve the target sample size in Tier 2 of 6,000. These children and young adults will complete both Tier 1 and Tier 2 assessments. See Table 2 for a full description of the promotion algorithm.

Children and young adults in the acute arm with a history of infection will also complete remote assessments at 2, 4, and 8 weeks after their infection onset, with additional in-person assessments at 8 weeks. Children and young adults in the acute arm without history of infection will complete the same assessments, timed relative to their enrollment date. All ABCD youth are eligible to participate in RECOVER Tier 1, and can be referred to a *de novo* cohort site to participate in Tiers 2 and 3, if geographically feasible.

*Tier 3* has the most clinically intensive assessments with longitudinal in-person visits for a subset at 12 and 24 months post-enrollment. Tier 3 will include 600 children and young adults with history of infection from Tier 2.

## Main categories of data

Data collected for the *de novo* and ABCD cohorts are described below (Table 3).

*Surveys* include validated surveys with NIH common data elements, as available, informed by expert opinion (S5 Table). All are completed using Research Electronic Data Capture (REDCap), with the child's first name coded within surveys to personalize the experience and to clarify which child the questions refer to given caregivers can have multiple children enrolled. For youth 17 years or younger, the caregiver is the primary respondent. Participants 18 through 25 years old are the primary respondent. Surveys assess sociodemographic information [38], child birth history [39], special health care needs [39–41], SARS-CoV-2 infection history, related conditions (e.g., MIS-C, POTS or other form of dysautonomia, and Long COVID diagnoses), COVID testing and vaccine history, COVID-related symptoms (both acute and long-term), COVID health consequences (e.g., diet [42], physical activity [42], sleep [42], screen time [42], schooling, parenting [43]) and social determinants of health (e.g., food insecurity [44], social support [45]). A list of potential Long COVID symptoms are assessed [18, 37] (Table 1), with respondents asked whether a specific problem or symptom is/was present for at least 4 weeks since the beginning of the COVID-19 pandemic and, for respondents with a history of infection, if the symptoms started before or after their infection.

*Clinical assessments* are completed at in-person Tier 2 visits across overarching domains of physical growth, physical health, neurocognition, and neurobehavioral function (S6 Table). Physical health domains include anthropometrics, vital signs, an active standing test measuring orthostatic blood pressures [46, 47], joint flexibility tests [48], electrocardiograms, and spirometry. Neurocognitive and neurobehavioral assessments vary by age (Table 4). Neurocognitive domains include broad and specific measures of attention, memory, receptive and expressive language skills, reading, and sensory function [49–53]. Neurobehavioral domains include a broad assessment of behavioral function including anxiety, mood, social interactions, aggression, sleep, self-regulatory behaviors, somatic complaints and attention concerns [54–61]. Tier 3 assessments follow the same domains, but provide more in-depth measurements. The promotion algorithm for Tier 3 is still under development. Physical health domains of cardio-pulmonary function are assessed by echocardiogram, cardiopulmonary exercise testing, cardiac MRI, pulmonary function tests, and sputum induction.

**Table 1. Potential post-acute sequelae of SARS-CoV-2 (PASC) symptoms being Assessed in RECOVER-Pediatrics\*.**

| Major or Minor Classification\*\* | Symptoms being Assessed |
|---|---|
| **General symptoms or problems** | |
| Major | Fever |
| | Feeling sleepy during the day[a] |
| | Fussy or cranky (crying a lot)[b] |
| | Low energy or not feeling strong enough to do things[a] |
| | Feeling very tired all day long[a] |
| | Feeling very tired after walking[a] |
| | Not wanting to eat (poor appetite) |
| | Lost weight or gained less than expected |
| | Lost height or grew less than expected |
| Minor | Trouble sleeping |
| | Hot and cold spells (feeling hot or cold for no reason)[a] |
| | Sweating more than normal |
| | Wanting to eat more than normal (increased appetite) |
| | Wanting to drink liquids more than normal (increased thirst) |
| | Gained weight more than expected |
| **Symptoms or problems in the eyes, ears, nose, and throat** | |
| Major | Light hurts your eyes[a] |
| | Change in hearing[a] |
| | Ringing in the ears[a] |
| | Change in smell[a] |
| | Loss of smell[a] |
| | Throat hurts (sore throat)[a] |
| | Loss of voice (sounding hoarse) |
| | Problems swallowing |
| | Change in how things taste[c] |
| Minor | Eyes look red |
| | Eyes are watery |
| | Eyes are dry |
| | Dark circles or color under the eyes |
| | Trouble seeing or blurry vision[a] |
| | Stuffy nose or runny nose |
| | Very dry mouth[a] |
| | Problems with teeth or gums |
| | Chapped lips |
| **Symptoms or problems involving the heart and lungs** | |
| Major | Dry cough |
| | Wet cough (brings up mucus) |
| | Barking cough |
| | Trouble breathing (breathing too fast) |
| | Pain when breathing[a] |
| | Pain in the chest[c] |
| | Feeling like your heart is beating really fast, racing, or pounding (called palpitations) when not doing exercise[c] |
| | Feeling like your heart is beating really fast when doing exercise[c] |
| | Fainting or feeling like you are going to faint (lightheaded) [c] |
| | Trouble walking[a] |
| | Trouble climbing stairs[a] |
| | Trouble running[a] |

(*Continued*)

**Table 1.** (Continued)

| Major or Minor Classification** | Symptoms being Assessed |
|---|---|
| **Symptoms or problems involving the belly** | |
| **Major** | Nausea (feeling like you are going to throw up) |
| | Throwing up (vomiting) |
| | Loose stool (diarrhea) |
| | Pain with peeing (urination)[a] |
| | Peeing more than normal (urination more than normal) [a] |
| **Minor** | Stomach pains/cramps[a] |
| | Trouble pooping/stooling (constipation) |
| **Symptoms or problems involving the skin, hair, and nails** | |
| **Major** | Skin rash |
| | Changes or problems with nails |
| | Changes or problems with hair |
| | Color changes in your skin, such as red, white or purple |
| | Color changes on the fingers or toes |
| **Minor** | Itchiness of the skin[a] |
| **Symptoms or problems involving the bones and muscles** | |
| **Major** | Muscle weakness |
| | Pains in the joints (like the elbows, knees, ankles)[a] |
| | Pain in the back[a] |
| | Pain in the neck[a] |
| **Minor** | Sore muscles or pain in the muscles[a] |
| | Body aches or pains |
| **Symptoms or problems involving the brain and nerves** | |
| **Major** | Headache[c] |
| | Feeling dizzy (feeling like the room is spinning)[c] |
| | Shakiness or tremors[c] |
| | Feeling tingling or 'pin-and-needles' in the hands and feet[c] |
| | Unable to move part of the body |
| | Problems with remembering things (memory)[a] |
| | Problems with focusing on things (concentration), sometimes called "brain fog" [a] |
| | Problems with talking [a] |
| **Symptoms or problems involving feelings or behavior** | |
| **Major** | Feeling sad or depressed[c] |
| | Feeling anxious or on edge[c] |
| | Feeling a lot of fear when being away from parent or caregiver[b] |
| | Feeling a lot of fear of specific things like spiders or being up high[d] |
| | Feeling a lot of fear about being with other children or adults[d] |
| | Feeling fear of crowds or being in closed-in spaces[c] |
| | Having a sudden intense feeling of fear, like a panic attack[e] |
| | Refusing to go to school[c] |
| | Seeing, hearing, or feeling that something is there when it is not (hallucinations)[c] |
| | Being hyperactive or much more active than other children[a] |
| | Refusing to follow rules or doing what they are asked to do[a] |
| | Serious breaking of rules like lying, stealing, starting fights, or bullying[a] |
| | Having repeating memories, dreams, thoughts, or worries after a traumatic event[a] |

(*Continued*)

**Table 1.** (Continued)

| Major or Minor Classification** | Symptoms being Assessed |
|---|---|
| **Minor** | Having a lot of tantrums[b] |
| | Holding their breath for a long time when they are afraid or angry[b] |
| | Having nightmares |
| | Screaming in fear while asleep, sometimes called night terrors[b] |
| | Aggressive behavior like hitting, biting or kicking[b] |
| | Rocking the body back and forth or head banging |
| **Symptoms or problems involving periods** | |
| **Minor** | Getting periods less often [f] |
| | Getting periods more often [f] |
| | Heavier periods [f] |
| | Lighter periods [f] |

* The following question is used to assess potential PASC symptoms in RECOVER-Pediatrics: "Did your child have any of these problems or symptoms lasting for more than 4 weeks that started or got worse since the COVID pandemic began in March 2020? These are problems or symptoms that kept happening without stopping or kept happening again and again for longer than 4 weeks."

** Symptoms were classified as major or minor depending on their severity and presumed likelihood of being associated with a COVID-19 infection.

[a] Symptoms included for children 3 years or older.

[b] Symptom included for children 5 years or younger.

[c] Symptom included for children 6 years or older.

[d] Symptoms included for children 2 years or older.

[e] Symptoms included for children 12 years or older.

[f] Symptoms included for those who report having menses.

Gastrointestinal function is assessed using abdominal ultrasound, and neurological function is assessed using brain MRI, electroencephalogram, and measures of neurocognitive function and psychiatric symptoms. These assessments include higher level measurement of all cognitive domains (thinking, language processing, memory, attention, and executive functioning) [62], visual motor integration and speed [63–65], and a psychiatric symptom battery [66].

*Biospecimens* are collected across all Tiers using kits designed specifically for each visit, timepoint, and participant age (Table 5; S6 Table). Tier 1 biospecimens consist of saliva and whole blood. Kits are shipped to homes for remote collection. Child and primary caregivers provide both saliva and blood; the other biological parent when available provides only saliva. Saliva is collected using Oragene devices (OGR-600) and banked for future DNA analysis. Whole blood is collected using a TASSO M20 device [67], which collects capillary blood using 4 volumetric sponges that each hold 17.5μL of blood (70 μL total). One sponge is used for SARS-CoV-2 spike and nucleocapsid antibody testing and remaining sponges are banked for future use.

Tier 2 acute biospecimens include saliva (Oragene OGR-600) and whole blood collections. All post-acute Tier 2 biospecimens consist of whole blood. The maximum amount of blood drawn at a single visit is age dependent. Whole blood is collected using serum separator tube (SST) and Ethylenediaminetetraacetic tube acid (EDTA) across all ages above 24 months and an additional cell preparation tube (CPT) is included for participants 6 years of age and older.

**Table 2. Promotion algorithm used in the *de novo* RECOVER-Pediatrics cohort for selecting children and young adults for the longitudinal follow-up (Tier 2).**

| Type of participant | Subgroup | Subgroup criteria | Promotion rate to Tier 2 |
|---|---|---|---|
| "*Acute infected*": Children and young adults who reported having a COVID infection ≤30 days prior to enrollment | All | None | 100% |
| "*Post-acute infected*": Children and young adults who reported having a COVID infection >30 days prior to enrollment | High probability of PASC[a] | Any of the following: <br>1. Prior diagnosis of Long COVID or MIS-C based on study site medical record review or referral by a health care provider <br>2. Recruitment from a Long COVID clinic <br>3. ≥1 fair/poor responses on the global health PROMIS scale[b] and ≥1 major or minor symptom reported[c] <br>4. ≥1 good/fair/poor responses on the global health PROMIS scale[b] and ≥ 1 major symptom reported[c] <br>5. ≥1 good/fair/poor responses on the global health PROMIS scale[b] and ≥2 minor symptoms reported[c] | 100% |
| | Medium probability of PASC[a] | Any of the following: <br>1. ≥1 good responses on the global health PROMIS scale[b] and ≥1 major or minor symptom reported[c] <br>2. ≥1 very good/good/fair/poor responses on the global health PROMIS scale[b] and ≥1 major symptom reported[c] <br>3. ≥1 very good/good/fair/poor responses on the global health PROMIS scale[b] and ≥2 minor symptoms reported[c] | 50% |
| | Low probability of PASC[a] | Does not meet high or medium probability of PASC criteria | 20% |
| "*Uninfected*": Children or Young adults without a known history of a COVID infection | Acute arm | At enrollment, 17% of the "uninfected" group were randomly assigned to participate in the acute arm of the study. | 100% |
| | Post-acute arm | At enrollment, 83% of the "uninfected" group were randomly assigned to participare in the post-acute arm of the study. 40% of this group will be randomly assigned to participate in Tier 2 | 40% |

[a] Responses to the PROMIS Global Health Scales and the presence of major and minor symptoms are used to categorize participants who are post-acute infected as high, medium, or low probability of PASC.

[b] The PROMIS Global Health Scales are self-reported or caregiver-reported measures of overall, physical, and mental health for young adults and children, respectively [34–36]. The three questions from the caregiver-reported version that are used in the algorithm, include: 1) "*In general, would you say your child's health is*?; 2) "*In general, how would you rate your child's physical health*?"; and 3) "*In general, how would you rate your child's mental health, including mood and ability to think*?" Responses include: *Excellent, Very Good, Good, Fair, or Poor.*

[c]A list of all major and minor symptoms, reported at the enrollment visit as part of the symptom screener, is provided in the Table 1 [18, 37]. Not all symptoms are asked of all participants, as many are age-specific (e.g., fewer symptoms assessed for younger children) and sex-specific (e.g., menses related symptoms).

Tier 3 biospecimens consist of whole blood, sputum, swabs (e.g., skin, nasal, oral), urine and stool. Collection of Tier 3 biospecimens is limited to children ages 3 years and older (maximum allowable volume is age dependent).

## Statistical methods

We will estimate the proportion of children and young adults experiencing new onset or worsening of each symptom (incidence), stratified by age (0–5, 6–12, 13–17, 18–25 years), over time. Age stratifications were based on child developmental stages, including early childhood (birth to 5 years), school-age (6 to 11 years), adolescence (12 to 17 years) and young adulthood (18 to 25 years) [39]. Prevalence within the recruited population will be estimated by calculating the point prevalence of each symptom by calculating the proportion of children and young adults who are currently experiencing each symptom at each study visit. The excess burden of

**Table 3. Summary of study assessments in the *de novo* RECOVER-Pediatrics cohort[a].**

Legend: Blue (B) = Questionnaires; Red (R) = Clinical Assessments; Yellow (Y) = Biospecimen Collection

| | BL | ... | 2 | 4 | 8 | ... | 2 to 6 | 12 | 24 | 36 | 48 |
|---|---|---|---|---|---|---|---|---|---|---|---|
| | | | (weeks) | | | | (months) | | | | |
| **Tier 1 and Tier 2 Assessments** | BL | | Acute phase | | | | Post-acute phase | | | | |
| Identity | B | | | | | | | | | | |
| Demographics | B | | | | | | B | B | B | B | B |
| Child Birth History | B | | | | B | | | | | | |
| Child Current Health Status | B | | | | | | B | B | B | B | B |
| Special Health Care Needs Screener | B | | | | | | B | B | B | B | B |
| PROMIS Global Health | B | | | | B | | B | B | B | B | B |
| First COVID Infection History | B | | | | | | | | | | |
| Current COVID Infection History | | | | | | | | | | | |
| Weekly COVID Infection History | | | B | B | B | | | | | | |
| COVID Infection History (Follow-up) | | | | | | | B | B | B | B | B |
| Related Conditions (MIS-C, POTS) | B | | | | | | B | B | B | B | B |
| COVID Testing History | B | | | | B | | | | | | |
| COVID Family Infection | B | | | | B | | B | B | B | B | B |
| COVID Symptoms | B | | B | B | B | | | | | | |
| COMPASS-31 | | | | | | | B | B | B | B | B |
| COVID Vaccine History | B | | | | B | | | | | | |
| COVID Health Consequences | B | | | | | | B | B | B | B | B |
| Social Determinants of Health | B | | | | | | B | B | B | B | B |
| Child Wellbeing | B | | | | B | | B | B | B | B | B |
| Tier 1 Biospecimen | Y | | | | | | | | | | |
| Anthropometry & Vital Signs | | | | | R | | R | R | R | R | R |
| Electrocardiogram | | | | | R | | R | R | R | R | R |
| Spirometry | | | | | R | | R | R | R | R | R |
| Pulse Oximetry | | | | | R | | | | | | |
| Tier 2 Acute Biospecimens | | | | | Y | | | | | | |
| 10 Minute Active Standing Test (Orthostatic BP) | | | | | | | R | R | R | R | R |
| Joint Flexibility (Beighton Scale) | | | | | | | R | R | R | R | R |
| Neurocognitive Development (e.g., NIH Toolbox) | | | | | | | R | R | R | R | R |
| Emotional/Mental Health | | | | | | | R | R | R | R | R |
| Post-Acute Tier 2 Biospecimens | | | | | | | Y | Y | Y | Y | Y |
| **Tier 3 Assessments** | | | | | | | | | | | |
| Echocardiogram | | | | | | | | R | R | | |
| Cardiac MRI | | | | | | | | R | R | | |
| Pulmonary Function Tests (PFTs) | | | | | | | | R | R | | |
| Lung Microbiome (Sputum Induction) | | | | | | | | R | R | | |
| Cardiopulmonary Exercise Testing | | | | | | | | R | R | | |
| Abdominal Ultrasound | | | | | | | | R | R | | |
| Brain MRI | | | | | | | | R | R | | |
| Brain EEG | | | | | | | | R | R | | |
| Neurocognitive Testing | | | | | | | | R | R | | |
| Tier 3 Biospecimens | | | | | | | | Y | Y | | |

[a]Blue = Questionnaires; Red = Clinical Assessments; Yellow = Biospecimen Collection

**Table 4. Neurocognitive, Neurobehavioral, Well-Being and mental health measures by age in Tiers 2 and 3 for the *de novo* RECOVER-Pediatrics cohort.**

| Study Tier | Neurocognitive and Developmental Assessments | Neurobehavioral, Well-Being and Mental Health Assessments |
|---|---|---|
| **Infancy and Toddlerhood: Birth through 2 years old** | | |
| Tier 2 | Ages and Stages Questionnaire—3rd Edition (ASQ-3) [49–51]<br>Modified Checklist for Autism in Toddlers Revised with Follow up (MCHAT-RF) [52] | Ages and Stages Questionnaires: Social-Emotional, 2nd Edition (ASQ:SE-2) [54]<br>Child Behavior Checklist [55] |
| Tier 3 | N/A | N/A |
| **Preschool-Age: 3 years old through 5 years old** | | |
| Tier 2 | Ages and Stages Questionnaire—3rd Edition (ASQ-3) [49–51]<br>NIH Toolbox Cognitive Measures [53] | Ages and Stages Questionnaires: Social-Emotional, 2nd Edition (ASQ:SE-2) [54]<br>Child Behavior Checklist [55]<br>Patient-Reported Outcomes Measurement Information System (PROMIS®) Parent Proxy Anger Scale [56]<br>PROMIS® Parent Proxy Psychological Stress Experiences Scale [57]<br>PROMIS® Parent Proxy Positive Affect Scale [58] |
| Tier 3 | *Cognitive*: Woodcock Johnson Cognitive Battery subtests [62]<br>*Language*: Woodcock-Johnson Oral Language Battery subtests [62]<br>*Verbal Memory*: Woodcock Johnson subtests [62]<br>*Visual Memory*: Wide Range Assessment of Memory and Learning [63]<br>*Visual-Motor Drawing*: Beery Buktenica [64]<br>*Visual Motor Speed*: Purdue Pegboard [65]<br>*Pre-Academics*: Woodcock-Johnson Achievement Battery subtests [62] | Kiddie SADS computer completed by caregiver [66] |
| **School-age and Adolescence: 6 years old through 17 years old** | | |
| Tier 2 | NIH Toolbox Cognitive Measures [53] | Patient-Reported Outcomes Measurement Information System (PROMIS®) Parent Proxy Anger Scale [56]<br>PROMIS® Parent Proxy Psychological Stress Experiences Scale [57]<br>PROMIS® Parent Proxy Positive Affect Scale [58]<br>Revised Children's Anxiety and Depression Scale (RCADS-25) [59]<br>Strengths and Difficulties Questionnaire (Hyperactivity/Inattention and Conduct Problems Subscales) [60] |
| Tier 3 | *Cognitive*: Woodcock Johnson Cognitive Battery subtests [62]<br>*Language*: Woodcock-Johnson Oral Language Battery subtests [62]<br>*Verbal Memory*: Woodcock Johnson subtests [62]<br>*Visual Memory*: Wide Range Assessment of Memory and Learning [63]<br>*Visual-Motor Drawing*: Beery Buktenica [64]<br>*Visual Motor Speed*: Purdue Pegboard [65]<br>*Pre-Academics*: Woodcock-Johnson Achievement Battery subtests [62] | Kiddie SADS computer completed by caregiver [66] |
| **Young Adults: 18 years through 25 years old** | | |
| Tier 2 | NIH Toolbox Cognitive Measures [53] | Patient-Reported Outcomes Measurement Information System (PROMIS®) Parent Proxy Anger Scale [56]<br>PROMIS® Parent Proxy Psychological Stress Experiences Scale [57]<br>Achenbach Adult Self Report [61] |
| Tier 3 | *Cognitive*: Woodcock Johnson Cognitive Battery subtests [62]<br>*Language*: Woodcock-Johnson Oral Language Battery subtests [62]<br>*Verbal Memory*: Woodcock Johnson subtests [62]<br>*Visual Memory*: Wide Range Assessment of Memory and Learning [63]<br>*Visual-Motor Drawing*: Beery Buktenica [64]<br>*Visual Motor Speed*: Purdue Pegboard [65] | SADS Structured Psychiatric Interview [66] |

each symptom due to infection will be assessed by calculating differences in incidence and prevalence between children and young adults with and without an infection history. Odds ratios and relative risks for the association between infection and onset of each symptom will also be calculated, adjusting for sex in each age strata. Logistic regression and poisson regression with robust standard errors will be used in these analyses [68].

**Table 5. Biospecimen collection and processing summary.**

| Participant Samples Collected For | Collected Specimen | Quantity[a] | Biobanked Specimen Type | Number of Aliquots | Aliquot Volume |
|---|---|---|---|---|---|
| **Tier 1** | | | | | |
| Pediatric Participant; Primary Caregiver; Additional Biologic Parent | OGR-600 –Saliva | 1 x 2mL | Saliva | NA | 2 mL |
| Pediatric Participant; Primary Caregiver | TASSO M20 –Capillary Blood | 1 x 70 µL | Blood | 4 x volumetric sponges | 17.5 µL |
| **Tier 2 Acute** | | | | | |
| All Age Groups over 24 months | Oragene 600 –Saliva | 1 x 2mL | Saliva | N/A | 2 mL |
| All Age Groups over 24 months | Serum Separator Tube (SST)–Whole Blood[b] | 1 x 5mL | Serum | 5 | 500 µL |
| All Age Groups over 24 months | EDTA–Whole Blood[c] | 1 x 10mL | • Plasma<br>• White Blood Cells (WBCs)<br>• Red Blood Cells (RBCs) | • 13 x Plasma<br>• 1 x WBC<br>• 3 x RBCs | • 5 x 200 µL Plasma (Rutgers)<br>• 8 x 500 µL Plasma (PBC)<br>• WBC– 1 mL<br>• RBCs– 1 mL |
| Ages 6–25 yrs. | Sodium Citrate Cell Preservation Tube (CPT)–Whole Blood[d] | 2 x 4 mL | • Peripheral Blood Mononuclear Cells (PBMCs) | ~3 x PBMCs (target cell count minimum 5 million cells/mL) | 1 mL |
| **Tier 2 Post Acute** | | | | | |
| Ages 24mo -under 6 years (all post-acute visits) | Serum Separator Tube (SST)–Whole Blood[b] | 1 x 5mL | Serum | 5 | • 500 µL |
| Ages 6–25 yrs (6 month visit only) | Serum Separator Tube (SST)–Whole Blood[b] | 2 x 5mL | Serum | 3 | • 1 x 1 mL—ARUP<br>• 1 x 2 mL—ARUP<br>• 1 x 1.5 mL–PBC<br>• 6 x 200 µL for Rutgers and/or<br>• 3 x 500 µL for PBC |
| Ages 6–9 yrs. | Sodium Citrate Cell Preservation Tube (CPT)–Whole Blood[d] | 2 x 4 mL | Peripheral Blood Mononuclear Cells (PBMCs) | 8 x PBMCs (target cell count minimum 5 million cells/mL) | 1 mL |
| Ages 10–25 yrs. | Sodium Citrate Cell Preservation Tube (CPT)–Whole Blood[d] | 4 x 4 mL | Peripheral Blood Mononuclear Cells (PBMCs) | 16 x PBMCs (target cell count minimum 5 million cells/mL) | 1 mL |
| All Age Groups EXCEPT 6-month Post Acute visit for age 6–9 yrs | EDTA–Whole Blood[c] | 1 x 10mL | • Plasma<br>• White Blood Cells (WBCs)<br>• Red Blood Cells (RBCs) | • 13 x Plasma<br>• 1 x WBC<br>• 3 x RBCs | • 5 x 200µL Plasma (Rutgers)<br>• 8 x 500µL Plasma (PBC)<br>• WBC– 1mL<br>• RBCs– 1 mL |
| **Tier 3[e]** | | | | | |
| All Age Groups | Serum Separator Tube (SST)–Whole Blood[b] | TBD | • Serum | • TBD | • TBD |

*(Continued)*

**Table 5.** (Continued)

| Participant Samples Collected For | Collected Specimen | Quantity[a] | Biobanked Specimen Type | Number of Aliquots | Aliquot Volume |
|---|---|---|---|---|---|
| All Age Groups | EDTA–Whole Blood[c] | TBD | • Plasma<br>• White Blood Cells (WBCs)<br>• Red Blood Cells (RBCs) | • TBD | • TBD |
| All Age Groups | Lithium Heparin–Whole Blood | TBD | • Plasma | • TBD | • TBD |
| All Age Groups | Red Top (No Additive)–Whole Blood | TBD | • Serum | • TBD | • TBD |
| All Age Groups | Other Biospecimens for Microbiome Analysis (e.g. sputum, swaps (skin, nasal, oral), urine, stool) | TBD | • Sputum<br>• Swabs<br>• Urine<br>• Stool | • TBD | • TBD |

[a]Sample volumes are age dependent: (newborn to under 6 years: maximum draw of 2 mL per kg of body weight; 6 to under 10 years: 25 mL; greater than 10 years old: 38 mL)

[b]SST tube is collected and within 4 hours of collection the SST tube is centrifuged, serum is aliquoted and frozen locally at collection sites. Serum aliquots are batch shipped frozen on dry ice in monthly intervals and are banked for future research.

[c]The EDTA tube is collected for all age groups and is processed for plasma, WBC, and RBC aliquots. A plasma aliquot is sent out for central testing. The other EDTA aliquot derivatives are frozen and banked for future research.

[d]The CPT tubes are only collected for age groups 6–25 years. The CPT tubes are centrifuged at collection sites and sent on ice packs day of collection to the PBC. Once arrived at the PBC, the CPT tubes are processed. A maximum of 8 x 1 mL PBMC aliquots (minimum of 5 million cells/mL) are derived. PBMC aliquots are stored in liquid nitrogen and banked for future research.

[e]Tier 3 biospecimen parameters are currently under development, but will involve collection of whole blood for clinical chemistry and biobanking and the collection and banking of biospecimens for microbiome analysis.

A preliminary working definition of PASC will be informed by using variable selection methods to identify which symptoms best differentiate children and young adults with and without an infection, following the methodology previously applied to develop a working definition of PASC in the RECOVER adult cohort [69]. Data from the Tier 1 visit will primarily be used. The estimated associations obtained from regression models will be used to define a PASC score, with a cutoff for PASC defined based on clinical expertise while ensuring that the rate of those with no history of infection who are diagnosed as having PASC is reasonably low. This preliminary symptom-based working definition is intended for research purposes and not clinical diagnosis. It will be modified and augmented by clinical and subclinical findings as they become available. The working definition of PASC that is developed will also be validated in RECOVER participants who have linked EHR data against other definitions derived from EHR-based cohorts [70]. While the working definition of PASC will be initially developed within each age strata, depending on the overlap of key symptoms that are identified that define PASC, some age groups may be aggregated in the interest of developing a more unified definition of PASC. To identify PASC phenotypes among children and young adults who are classified as having PASC defined by symptom patterns, we will use unsupervised learning methods to discover symptom clusters within each age strata (e.g., agglomerative hierarchical clustering [71] and consensus clustering [72]) to define PASC sub-phenotypes.

With this definition of PASC, we will conduct regression analyses to evaluate whether the risk of PASC and PASC sub-types differs by multiple factors, including demographic, clinical, and caregiver characteristics, social determinants of health, SARS-CoV-2 infection and immunization history, symptom severity during the acute phase of SARS-CoV-2 infection, and

therapeutic exposures. Logistic and Poisson regression will be used to evaluate the association (i.e., odds ratios and risk ratios) between pre-infection factors and PASC as a binary outcome, and multinomial regression will be used when PASC sub-types are used as categorical outcomes. Among participants in Tier 2 who develop PASC, we will use time-to-event analyses (i.e., Cox proportional hazards regression) to identify factors that influence time to recovery from PASC. To investigate biomarkers related to PASC, clinical laboratory assessments will be compared between children and young adults who do and do not develop PASC. Mediation analyses will also be used to study the pathways by which SARS-CoV-2 infection leads to the development of PASC. Since the trajectory of how PASC manifests may be affected by the presence of pre-existing conditions, we will study separately the pathophysiology of PASC in children and young adults with and without such conditions, when appropriate.

The study aims cover a wide range of scientific questions, but not all analyses will involve hypothesis testing. For instance, defining PASC does not require hypothesis testing, but evaluating whether the definition of or rates of PASC differ between age groups does. When multiple comparisons are made across age groups or other defined strata, or when different exposures and outcomes are assessed within the same subgroups, multiplicity adjustments for testing of non-exploratory hypotheses will be performed using the Hochberg procedure, in which tests of significance are performed in order of decreasing p-value with increasingly stringent thresholds [73]. This approach has been found to sacrifice less power in observational studies with correlated outcomes compared to the Bonferroni and other standard approaches for addressing multiplicity [74].

Potential sources of missing data include item nonresponse and attrition. Multiple imputation by chained equations will be the primary approach used to handle item nonresponse [75]. Sensitivity analyses will include adjustments for potentially missing not at random data (i.e., informative missingness) using pattern mixture models, which permit the distribution of missing variables to differ between observed and unobserved values. Attrition, or missed visits, in the longitudinal phase of the study (Tier 2) will be addressed depending on the affected analysis. For time-to-event modeling, attrition induces censoring, which may not be independent if participants drop out of the study in a systematic fashion (i.e., participants with worse symptom trajectories may be less likely to continue to participate in the study). Inverse probability of censoring weights will be used to address dependent censoring in this context [76]. For analyses with repeated measures, multiple imputation alongside likelihood-based methods, which are robust to the missing at random assumption, [77] will be used to handle missing data, with pattern mixture models used to perform sensitivity analyses for informatively missing data as appropriate [78].

Statistical analyses will primarily be conducted in R and SAS, though the statistical packages used by individual investigators for future analyses beyond those described in this manuscript will vary.

## Power calculations

Power calculations for the *de novo cohort* were performed prior to recruitment using a type 1 error rate of 0.01 as a preliminary multiplicity adjustment. The actual statistical approach for addressing multiplicity will differ depending on the analysis and will involve more sophisticated methods (see Statistical methods), but these are not amenable to most power calculations. With 4,800 infected and 1,200 uninfected children and young adults from both acute and post-acute arms in Tier 1, as well as 10,000 children ages 12–17 from the ABCD cohort (assuming 3,500 are infected and 6,500 are uninfected), assuming the risk of a given symptom in the uninfected group is 10%, we have 90% power to detect a difference as small as 1.9% in the frequency of that symptom between groups.

In Tier 2, given the sampling and promotion framework described in *Timing of Study Assessments*, our sample with longitudinal follow-up will be skewed towards those who are more likely to have PASC. Following development of a definition of PASC, we consider the scenario in which we assume that of the 5,400 children and young adults with a history of infection in Tier 2, 3,600 meet PASC criteria and 1,800 do not. For a hypothetical risk factor with 50% prevalence in the PASC- group, we have 90% power to detect an odds ratio as small as 1.25 for the odds of PASC for those with the risk factor versus those without. For a factor with 25% prevalence in the PASC-negative group, the minimum detectable odds ratio is 1.28. In our Tier 3 sample of 600 children and young adults with history of infection (which includes additional data on biomarkers), assuming the sample has 400 with PASC and 200 without PASC, for a marker with 10% prevalence in the PASC- group, we have 90% power to detect an odds ratio as small as 2.60 for PASC.

Given that many analyses will be stratified by age group, we calculate minimum detectable effect sizes overall and within each age stratum in S7 Table. Estimates of the distribution of ages are based on early enrollment data, with 26% of main cohort participants in the age 0–5 years category, 28% in ages 6–11 years, 26% in ages 12–17 years, and 20% in ages 18–25 years.

## Discussion

The overall goal of RECOVER-Pediatrics is to improve our understanding of recovery after SARS-CoV-2 infection, with a focus on the prevalence, natural history, and pathogenesis of PASC in children and young adults. Successful completion should lead to formal characterization of pediatric PASC as its own syndrome. This is essential to develop diagnostic, treatment, and preventive strategies tailored to children's unique physiology.

RECOVER-Pediatrics is well positioned to ascertain the epidemiology, four-year clinical course, and sociodemographic contributions to pediatric PASC, with rich data and biosamples available to readily test further mechanistic hypotheses, establish biomarkers, and provide insights into potential therapies. The meta-cohort is designed to provide details that are not available in other large epidemiologic or electronic health records queries, including a dynamic study design that can be flexible and responsive as new variants arise, and as our understanding of the long-term effects of SARS-CoV-2 evolves. RECOVER-Pediatrics was designed to include a wide range of ages, and diverse socioeconomic, racial, ethnic and geographic populations to ensure that findings are generalizable, and provide equitable benefit for all.

The generation-defining nature of the COVID-19 pandemic will impact the life course of children in ways that we have yet to fully understand. The unprecedented scope of RECOVER-Pediatrics sets the stage for not only characterizing a new disorder that will impact children for years to come, but also for identifying and deploying solutions through its collaborations with investigators and communities across the country.

RECOVER-Pediatrics is expected to gather a rich data set that can be used to develop treatments for persons with Long COVID and provide guidelines for how to respond more quickly to prevent, reduce the consequences, and treat complications of future coronavirus outbreaks which are likely to emerge.

## Supporting information

**S1 Checklist. SPIRIT clinical trials checklist.**
(DOCX)

**S1 Table. Hubs and enrolling sites.**
(DOCX)

**S2 Table. Inclusion and exclusion criteria.**
(DOCX)

**S3 Table. Inclusion into analytic groups.**
(DOCX)

**S4 Table. World Health Organization (WHO) criteria.**
(DOCX)

**S5 Table. Survey topics in tiers 1 and 2 questionnaires.**
(DOCX)

**S6 Table. Clinical and laboratory assessments across the tiers in the de novo RECOVER-Pediatrics cohort.**
(DOCX)

**S7 Table. Power calculations to determine minimum detectable effect sizes, stratified by age group.**
(DOCX)

**S1 Appendix. RECOVER-Pediatrics consortium members.**
(DOCX)

**S1 Protocol. RECOVER-Pediatrics protocol.**
(PDF)

**S1 Text.**
(PDF)

**S2 Text.**
(PDF)

**S3 Text.**
(PDF)

**S4 Text.**
(PDF)

**S5 Text.**
(PDF)

**S6 Text.**
(PDF)

**S1 File.**
(PDF)

**S2 File.**
(PDF)

**S3 File.**
(PDF)

**S4 File.**
(PDF)

**S5 File.**
(PDF)

**S6 File.**
(PDF)

**S7 File.**
(PDF)

**S1 Data.**
(ZIP)

## Acknowledgments

We would like to thank the National Community Engagement Group (NCEG), all patient, caregiver and community representatives, and all the participants enrolled in the RECOVER initiative.

**Disclaimer:** Authorship has been determined according to ICMJE recommendations. The content is solely the responsibility of the authors and does not necessarily represent the official views of the RECOVER program, the NIH or other funders.

## Author Contributions

**Conceptualization:** Rachel S. Gross, Tanayott Thaweethai, Erika B. Rosenzweig, Lori B. Chibnik, Valerie J. Flaherman, Andrea S. Foulkes, Richard Gallagher, Terry L. Jernigan, Elizabeth W. Karlson, Stuart D. Katz, Patricia A. Kinser, Lawrence C. Kleinman, Michelle F. Lamendola-Essel, Joshua D. Milner, Sindhu Mohandas, Jane W. Newburger, Kyung E. Rhee, Melissa S. Stockwell, Kelan G. Tantisira, Moriah E. Thomason, David Warburton, John C. Wood, Andrew M. Atz, Robin L. Aupperle, Fiona C. Baker, Deanna M. Barch, Hülya Bükülmez, Rebecca G. Clifton, Viren D'Sa, E. Vincent S. Faustino, Laura Hobart-Porter, Matthew Huentelman, Vanessa L. Jacoby, David C. Kaelber, Alan L. Mendelsohn, Torri D. Metz, Michael C. Neale, Matthew E. Oster, Hengameh Raissy, Jacqueline Szmuszkovicz, William T. Zempsky.

**Data curation:** Tanayott Thaweethai, James Chan, Andrea S. Foulkes, Stuart D. Katz, Shifa Ahmed, Arielle Baskin-Sommers, Marie-Abele C. Bind, Walter Dehority, John J. Foxe, Kevin M. Gray, Ashraf S. Harahsheh, Daniel S. Hsia, Joanna Jacobus, Tyler J. Kasmarcak, Angela R. Laird, Peter Paul C. Lim, Brian W. McCrindle, Sheila M. Nolan, Harrison T. Reeder, Mark W. Russell, Alice I. Sato, Lindsay M. Squeglia, Maria M. Talavera-Barber, Ronald J. Teufel, II, Emily Zimmerman.

**Formal analysis:** James Chan, Lori B. Chibnik, David Warburton, Mark W. Russell.

**Funding acquisition:** Rachel S. Gross, Tanayott Thaweethai, Amy J. Elliott, Andrea S. Foulkes, Terry L. Jernigan, Stuart D. Katz, Lawrence C. Kleinman, Kyung E. Rhee, Melissa S. Stockwell, David Warburton, John C. Wood, Robin L. Aupperle, Fiona C. Baker, Duncan B. Clark, Alexander G. Fiks, John J. Foxe, Dylan G. Gee, Christina M. Hester, Matthew Huentelman, Krista M. Lisdahl, Russell J. McCulloh, Torri D. Metz, Eva M. Müller-Oehring, Michael C. Neale, Alice I. Sato, Rangaraj Selvarangan, Emily Zimmerman.

**Investigation:** Rachel S. Gross, Erika B. Rosenzweig, Valerie J. Flaherman, Lawrence C. Kleinman, Sindhu Mohandas, Kyung E. Rhee, Melissa S. Stockwell, Kelan G. Tantisira, Dongngan T. Truong, David Warburton, John C. Wood, Judy L. Aschner, Andrew M. Atz, Venkataraman Balaraman, Dithi Banerjee, Deanna M. Barch, Amanda L. Bogie, Elliott

Bueler, Hülya Bükülmez, B. J. Casey, Linda Chang, Lesley Cottrell, Viren D'Sa, Walter Dehority, Kirsten B. Dummer, Matthew D. Elias, E. Vincent S. Faustino, Alexander G. Fiks, Daniel Forsha, John J. Foxe, Sunanda Gaur, Dylan G. Gee, Kevin M. Gray, Mary M. Heitzeg, Christina M. Hester, Sophia Hill, Travis K. F. Hong, Carol R. Horowitz, Daniel S. Hsia, Katherine Irby, Joanna Jacobus, Vanessa L. Jacoby, David C. Kaelber, Matthew J. Kluko, Jessica S. Kosut, Jeremy Landeo-Gutierrez, Sean M. Lang, Peter Paul C. Lim, Krista M. Lisdahl, Russell J. McCulloh, Torri D. Metz, Manette Ness-Cochinwala, Matthew E. Oster, R. Mark Payne, Hengameh Raissy, Suchitra Rao, Mark W. Russell, Arash A. Sabati, Alice I. Sato, Rangaraj Selvarangan, Lindsay M. Squeglia, Jacqueline Szmuszkovicz, Maria M. Talavera-Barber, Deepika Thacker, Alan Werzberger, H. Shonna Yin, William T. Zempsky, Emily Zimmerman.

**Methodology:** Rachel S. Gross, Tanayott Thaweethai, Erika B. Rosenzweig, Lori B. Chibnik, Amy J. Elliott, Valerie J. Flaherman, Andrea S. Foulkes, Richard Gallagher, Elizabeth W. Karlson, Stuart D. Katz, Patricia A. Kinser, Lawrence C. Kleinman, Michelle F. Lamendola-Essel, Joshua D. Milner, Sindhu Mohandas, Jane W. Newburger, Kyung E. Rhee, Amy L. Salisbury, Cheryl R. Stein, Melissa S. Stockwell, Moriah E. Thomason, David Warburton, Shifa Ahmed, Robin L. Aupperle, Venkataraman Balaraman, Marie-Abele C. Bind, Andrew C. Heath, Carol R. Horowitz, Matthew Huentelman, Joanna Jacobus, Vanessa L. Jacoby, Krista M. Lisdahl, Alan L. Mendelsohn, Torri D. Metz, Erica R. Nahin, Michael C. Neale, Carlos R. Oliveira, Hengameh Raissy, Isabelle G. Randall, Harrison T. Reeder, Johana M. Rosas, Lindsay M. Squeglia, Michelle D. Stevenson, Maria M. Talavera-Barber, Ronald J. Teufel, II, Megan R. Warner, Jordan C. Weyer, H. Shonna Yin, William T. Zempsky.

**Project administration:** Rachel S. Gross, Tanayott Thaweethai, Erika B. Rosenzweig, James Chan, Mine S. Cicek, Andrea S. Foulkes, Terry L. Jernigan, Stuart D. Katz, Lawrence C. Kleinman, Michelle F. Lamendola-Essel, Praveen C. Mudumbi, Kyung E. Rhee, Melissa S. Stockwell, David Warburton, John C. Wood, Shifa Ahmed, Akram N. Alshawabkeh, Brett R. Anderson, Andrew M. Atz, Robin L. Aupperle, Fiona C. Baker, Dithi Banerjee, Arielle Baskin-Sommers, Sultana Bhuiyan, Marie-Abele C. Bind, Amanda L. Bogie, Natalie C. Buchbinder, Elliott Bueler, Hülya Bükülmez, Linda Chang, Duncan B. Clark, Rebecca G. Clifton, Katharine N. Clouser, Walter Dehority, Shari Esquenazi-Karonika, Danielle N. Evans, John J. Foxe, Naomi P. Friedman, Dylan G. Gee, Kevin M. Gray, Andrew C. Heath, Christina M. Hester, Sophia Hill, Daniel S. Hsia, Kathy D. Hummel, Katherine Irby, Joanna Jacobus, David C. Kaelber, Christine L. Larson, Krista M. Lisdahl, Russell J. McCulloh, Torri D. Metz, Eva M. Müller-Oehring, R. Mark Payne, Hengameh Raissy, Harrison T. Reeder, Alice I. Sato, Divya Shakti, Kavita Sharma, Lindsay M. Squeglia, Michelle D. Stevenson, Ronald J. Teufel, II, Mmekom M. Udosen, Megan R. Warner, Sara E. Watson, Jordan C. Weyer, Marion J. Wood, H. Shonna Yin.

**Resources:** Tanayott Thaweethai, Mine S. Cicek, Andrea S. Foulkes, Stuart D. Katz, Sindhu Mohandas, Shifa Ahmed, Akram N. Alshawabkeh, Judy L. Aschner, Arielle Baskin-Sommers, Sultana Bhuiyan, Marie-Abele C. Bind, Mirella Dapretto, Walter Dehority, Danielle N. Evans, Andrew C. Heath, Mary M. Heitzeg, Matthew Huentelman, Joanna Jacobus, David C. Kaelber, Russell J. McCulloh, Manette Ness-Cochinwala, Harrison T. Reeder, Mark W. Russell, Michael S. Schechter, Lindsay M. Squeglia, Mmekom M. Udosen, H. Shonna Yin, Emily Zimmerman.

**Software:** Tanayott Thaweethai, James Chan, Andrea S. Foulkes, Shifa Ahmed, Marie-Abele C. Bind, Harrison T. Reeder.

**Supervision:** Rachel S. Gross, Tanayott Thaweethai, Valerie J. Flaherman, Andrea S. Foulkes, Terry L. Jernigan, Stuart D. Katz, Michelle F. Lamendola-Essel, Kyung E. Rhee, Melissa S. Stockwell, Kelan G. Tantisira, David Warburton, Brett R. Anderson, Andrew M. Atz, Robin L. Aupperle, Dithi Banerjee, Arielle Baskin-Sommers, Marie-Abele C. Bind, Amanda L. Bogie, Elliott Bueler, Hülya Bükülmez, Linda Chang, Duncan B. Clark, Lesley Cottrell, Mirella Dapretto, Walter Dehority, Danielle N. Evans, John J. Foxe, Dylan G. Gee, Ashraf S. Harahsheh, Mary M. Heitzeg, Christina M. Hester, Sophia Hill, Daniel S. Hsia, Kathy D. Hummel, Katherine Irby, Joanna Jacobus, Vanessa L. Jacoby, David C. Kaelber, Krista M. Lisdahl, Russell J. McCulloh, Eva M. Müller-Oehring, Manette Ness-Cochinwala, Suchitra Rao, Harrison T. Reeder, Mark W. Russell, Yamuna Sanil, Michael S. Schechter, Rangaraj Selvarangan, Divya Shakti, Lindsay M. Squeglia, Michelle D. Stevenson, Maria M. Talavera-Barber, Ronald J. Teufel, II, Mmekom M. Udosen, Megan R. Warner, Sara E. Watson, Alan Werzberger, Jordan C. Weyer, H. Shonna Yin.

**Validation:** James Chan, Walter Dehority, Ashraf S. Harahsheh, Suchitra Rao, Alice I. Sato.

**Writing – original draft:** Rachel S. Gross, Tanayott Thaweethai, Erika B. Rosenzweig, Lori B. Chibnik, Mine S. Cicek, Richard Gallagher, Patricia A. Kinser, Lawrence C. Kleinman, Michelle F. Lamendola-Essel, Sindhu Mohandas, Praveen C. Mudumbi, Kyung E. Rhee, Cheryl R. Stein, Melissa S. Stockwell, Kelan G. Tantisira, Moriah E. Thomason, David Warburton, Torri D. Metz, Hengameh Raissy, Benard P. Dreyer.

**Writing – review & editing:** Rachel S. Gross, Tanayott Thaweethai, Erika B. Rosenzweig, James Chan, Lori B. Chibnik, Mine S. Cicek, Amy J. Elliott, Valerie J. Flaherman, Andrea S. Foulkes, Margot Gage Witvliet, Richard Gallagher, Maria Laura Gennaro, Terry L. Jernigan, Elizabeth W. Karlson, Stuart D. Katz, Patricia A. Kinser, Lawrence C. Kleinman, Michelle F. Lamendola-Essel, Sindhu Mohandas, Praveen C. Mudumbi, Jane W. Newburger, Kyung E. Rhee, Amy L. Salisbury, Jessica N. Snowden, Cheryl R. Stein, Melissa S. Stockwell, Kelan G. Tantisira, Moriah E. Thomason, David Warburton, John C. Wood, Shifa Ahmed, Almary Akerlundh, Akram N. Alshawabkeh, Brett R. Anderson, Judy L. Aschner, Andrew M. Atz, Robin L. Aupperle, Fiona C. Baker, Dithi Banerjee, Deanna M. Barch, Marie-Abele C. Bind, Tamara Bradford, Linda Chang, Maryanne Chrisant, Rebecca G. Clifton, Katharine N. Clouser, Lesley Cottrell, Kelly Cowan, Viren D'Sa, Soham Dasgupta, Walter Dehority, Audrey Dionne, Matthew D. Elias, Shari Esquenazi-Karonika, E. Vincent S. Faustino, Alexander G. Fiks, Daniel Forsha, John J. Foxe, Greta Fry, Sunanda Gaur, Dylan G. Gee, Kevin M. Gray, Stephanie Handler, Keren Hasbani, Camden Hebson, Mary M. Heitzeg, Christina M. Hester, Laura Hobart-Porter, Carol R. Horowitz, Daniel S. Hsia, Kathy D. Hummel, Katherine Irby, Vanessa L. Jacoby, Pei-Ni Jone, David C. Kaelber, Peter Paul C. Lim, Krista M. Lisdahl, Brian W. McCrindle, Russell J. McCulloh, Kimberly McHugh, Torri D. Metz, Julie Miller, Elizabeth C. Mitchell, Lerraughn M. Morgan, Eva M. Müller-Oehring, Erica R. Nahin, Michael C. Neale, Sheila M. Nolan, Carlos R. Oliveira, Onyekachukwu Osakwe, Matthew E. Oster, Michael A. Portman, Hengameh Raissy, Suchitra Rao, Harrison T. Reeder, Johana M. Rosas, Mark W. Russell, Arash A. Sabati, Yamuna Sanil, Alice I. Sato, Rangaraj Selvarangan, S. Kristen Sexson Tejtel, Divya Shakti, Lindsay M. Squeglia, Shubika Srivastava, Michelle D. Stevenson, Jacqueline Szmuszkovicz, Maria M. Talavera-Barber, Ronald J. Teufel, II, Felicia Trachtenberg, Sara E. Watson, Jordan C. Weyer, Marion J. Wood, H. Shonna Yin, William T. Zempsky, Emily Zimmerman, Benard P. Dreyer.

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
