## [Decision Letter · Decision Letter 0]

6 Jul 2023

PONE-D-23-10495Researching COVID to enhance recovery (RECOVER) pediatric study protocol: Rationale, objectives and designPLOS ONE

Dear Dr. Gross,

Thank you for submitting your manuscript to PLOS ONE. After careful consideration, we feel that it has merit but does not fully meet PLOS ONE’s publication criteria as it currently stands. Therefore, we invite you to submit a revised version of the manuscript that addresses the points raised during the review process.

We look forward to receiving your revised manuscript.

Kind regards,

Seyed Aria Nejadghaderi

Academic Editor

PLOS ONE

Journal Requirements:

2. Please provide additional details regarding participant consent. Specifically, please clarify how consent will be collected from the parents or legal guardians of the minors included in your study. In the ethics statement in the Methods and online submission information, please ensure that you have specified what type of consent you obtained form the parents or legal guardians (for instance, written or verbal, and if verbal, how it was documented and witnessed).

“I have read the journal's policy and the authors of this manuscript have the following competing interests: Brett Anderson reported receiving direct support for work not related to RECOVER work/publications from Genentech and the National Institute of Allergy and Immunology.

Walter Dehority reported receiving grant support from Merck and participating in research for the Moderna COVID-19 pediatric vaccine trial and the Pfizer Paxlovid trial.

Alex Fiks reported receiving support from NJM insurance and personal consulting fees not related to this paper from Rutgers University and the American Academy of Pediatrics.

Ashraf Harahsheh reported serving as a scientific advisory board member unrelated to this paper for OP2 DRUGS.

Lawrence Kleinman reported serving as an unpaid member of the Board of Directors for the DARTNet Institute, as a principle investigator at Quality Matters, Inc., and as the Vice Chair for the Borough of Metuchen Board of Health. Dr. Kleinman also reported grant support for work not related to RECOVER work/publications from NIH, HRSA, and the Robert Wood Johnson Foundation. Dr. Kleinman also reported minority individual stock ownership in Apple Computer, Sanofi SA, Experion, GlaxoSmithKline, Magyar Bank, Regeneron Pharmaceuticals, JP Morgan Chase, and Amgen Inc.

Torri Metz reported participating as a Principle Investigator in the medical advisory board for the planning of a Pfizer clinical trial of SARS-CoV-2 vaccination in pregnancy. She is also a principle investigator for a Pfizer study evaluating the pharmacokinetics of Paxlovid in pregnant people with COVID-19.

Joshua Milner reported serving as a member of the Scientific Advisory Board for Blueprint Medicines, in a capacity unrelated to RECOVER work/publications.”

4. One of the noted authors is a group or consortium “RECOVER Initiative”. In addition to naming the author group, please list the individual authors and affiliations within this group in the acknowledgments section of your manuscript. Please also indicate clearly a lead author for this group along with a contact email address.

Reviewers' comments:

Reviewer's Responses to Questions

**Comments to the Author**

1. Does the manuscript provide a valid rationale for the proposed study, with clearly identified and justified research questions?

Reviewer #1: Yes

Reviewer #2: Yes

Reviewer #3: Yes

Reviewer #4: Yes

2. Is the protocol technically sound and planned in a manner that will lead to a meaningful outcome and allow testing the stated hypotheses?

Reviewer #1: Yes

Reviewer #2: Yes

Reviewer #3: Yes

Reviewer #4: Yes

3. Is the methodology feasible and described in sufficient detail to allow the work to be replicable?

Reviewer #1: Yes

Reviewer #2: Yes

Reviewer #3: Yes

Reviewer #4: Yes

4. Have the authors described where all data underlying the findings will be made available when the study is complete?

Reviewer #1: Yes

Reviewer #2: No

Reviewer #3: Yes

Reviewer #4: Yes

5. Is the manuscript presented in an intelligible fashion and written in standard English?

Reviewer #1: Yes

Reviewer #2: Yes

Reviewer #3: Yes

Reviewer #4: Yes

6. Review Comments to the Author

You may also provide optional suggestions and comments to authors that they might find helpful in planning their study.

Reviewer #1: I read the protocol with interest and have no major comment. There is a need for such large-scale studies to present data in a meta fashion.

My decision is to accept it; however, I wish to ask the authors how they plan to deal with pediatric patients with pre-existing diseases, especially those with autoimmune disorders and inborn errors of immunity. Then, I suggest to include it briefly in methods or discussion.

Reviewer #2: How the age groups were determined for the study sample?

Power calculation: Effect sizes need to be justified. Since most of the analyses appear to be stratified by age groups, it is important to consider the power calculation for each stratum. Given the number of variables being tested, a more stringent Type I error may be considered.

How do you plan to address the issue of multiple comparisons and adjust for multiplicity?

The determination of PASC is not clearly written. How will you validate the definition of PASC? What if the PASC cannot be differentiated, especially the sample size may be small in each age strata?

Statistical Analysis:

Which statistical models do you plan to use to address the research questions? Logistic regression? Multiple regression? Will you use models for repeated measurements or survival models for time-to-event data?

Reviewer #3: Overall, this article provides a comprehensive overview of the RECOVER-Pediatrics study, which aims to characterize the clinical course, underlying mechanisms, and long-term health effects of post-acute sequelae of SARS-CoV-2 (PASC) in children and young adults. The article effectively highlights the unique challenges in understanding PASC symptoms in children and emphasizes the need for large-scale studies to define and recognize PASC in this population. The study design and methodology are well-described, and the article provides a clear outline of the study aims and objectives.

Minor comments:

The article would benefit from a clearer and more concise introduction. The current introduction provides a lot of statistics and information upfront, which can be overwhelming for the reader.

Consider restructuring the introduction to provide a brief overview of PASC in children and its impact before delving into the prevalence and incidence statistics.

Some of the sentences in the article are long and complex, which can make it difficult to follow the main points. Try breaking down these sentences into smaller, more digestible chunks to improve readability.For example, in the sentence "Given that young children might not be able to articulate symptoms, studies must rely on caregiver interpretation," breaking it into two sentences would make it easier to follow.

It would be helpful to provide more information about the selection process and criteria for the 10 hubs managing the study sites. Additionally, mentioning the geographical distribution of the sites or regions covered would provide context and a better understanding of the study's reach.

Reviewer #4: In this manuscript, Gross and colleagues present the protocol of the RECOVER-Pediatrics study, which aims to comprehensively characterize the clinical course, underlying mechanisms, and long-term effects of post-acute sequelae of SARS-CoV-2 from birth through 25 years old. Overall, the manuscript effectively introduces a well-designed protocol. However, I have a few suggestions for improvement:

1- The authors should consider adding a specific definition of post-acute sequelae of SARS-CoV-2 (PASC) in the first paragraph of the introduction. Providing a concise and clear definition would enhance the readers' understanding of the study's focus and objectives.

2- It would be beneficial if the authors could elaborate on how they plan to handle missing data. Addressing this aspect in the manuscript would enhance the transparency of the study's methodology.

3- In order to provide a more comprehensive understanding of the statistical analysis plan, the authors should consider including additional details, such as the software that will be used for the analysis and the specific statistical tests that will be employed. This information would help readers evaluate the rigor and validity of the study's statistical approach.

7. PLOS authors have the option to publish the peer review history of their article (what does this mean?). If published, this will include your full peer review and any attached files.

Reviewer #1: No

Reviewer #2: No

Reviewer #3: No

Reviewer #4: No

---

## [Author Response · Author response to Decision Letter 0]

30 Oct 2023

Reviewer #1: 

1. I read the protocol with interest and have no major comment. There is a need for such large-scale studies to present data in a meta fashion. My decision is to accept it; however, I wish to ask the authors how they plan to deal with pediatric patients with pre-existing diseases, especially those with autoimmune disorders and inborn errors of immunity. Then, I suggest to include it briefly in methods or discussion.

Thank you for this comment. We agree that is important to collect data about pre-existing diseases. We added information to the methods describing our assessment of special health care needs in children and young adults. We also added details in the statistical analysis section about how we can account for special health care needs in the analyses. For example, certain analyses can exclude participants with certain conditions or adjust for differences statistically as appropriate. Furthermore, additional analyses will focus on certain conditions in order to learn about Long COVID in children with these specific special health care needs. 

The following text in italics was added to the manuscript in the section called Main categories of data:

“Surveys assess sociodemographic information [36], child birth history [37], special health care needs [37-39], including an assessment of pre-existing conditions, SARS-CoV-2 infection history, related conditions (e.g., MIS-C, POTS or other form of dysautonomia, and Long COVID diagnoses), COVID testing and vaccine history, COVID-related symptoms (both acute and long-term), COVID health consequences (e.g., diet [40], physical activity [40], sleep [40], screen time [40], schooling, parenting [41]) and social determinants of health (e.g., food insecurity [42], social support [43]).”

The following text was added to the manuscript in the statistical methods section:

“Logistic and poisson regression will be used to evaluate the association (i.e., odds ratios and risk ratios) between pre-infection factors and PASC as a binary outcome, and multinomial regression will be used when PASC sub-types are used as categorical outcomes.”

The following text was also added to the manuscript in the statistical methods section:

“Since the trajectory of how PASC manifests may be affected by the presence of pre-existing conditions, we will study separately the pathophysiology of PASC in children and young adults with and without such conditions, when appropriate.” 

Reviewer #2: 

2. How the age groups were determined for the study sample?

RECOVER-Pediatrics was designed to study Long COVID across the entire childhood life course. Age stratifications were based on child developmental stages, including early childhood (birth to 5 years), school-age (6 to 11 years), adolescence (12 to 17 years) and young adulthood (18 to 25 years). The age groups parallel the age grouping used in the National Survey of Children’s Health. We chose to study these different developmental stages to determine if PASC will need to be defined differently for different age groups. 

To clarify this for the reader, the following text was added to the first paragraph of the statistical methods section:

“Age stratifications were based on child developmental stages, including early childhood (birth to 5 years), school-age (6 to 11 years), adolescence (12 to 17 years) and young adulthood (18 to 25 years).”

3. Power calculation: Effect sizes need to be justified. Since most of the analyses appear to be stratified by age groups, it is important to consider the power calculation for each stratum. Given the number of variables being tested, a more stringent Type I error may be considered. 

Thank you for this comment. We agree that more details about power is needed given that we will be conducting stratified analyses by age groups, since PASC may present differently for different age groups.

The following text was added to the first paragraph of the statistical methods section:

“Age stratifications were based on child developmental stages, including early childhood (birth to 5 years), school-age (6 to 11 years), adolescence (12 to 17 years) and young adulthood (18 to 25 years).”

The following text was added to the second paragraph of the statistical methods section:

“While the working definition of PASC will be initially developed within each age strata, depending on the overlap of key symptoms that are identified that define PASC, some age groups may be aggregated in the interest of developing a more unified definition of PASC.”

The following text was added to the end of the power calculations section:

“Given that many analyses will be stratified by age group, we calculate minimum detectable effect sizes overall and within each age stratum in Supplemental Table 7. Estimates of the distribution of ages are based on early enrollment data, with 26% of main cohort participants in the age 0-5 year category, 28% in ages 6-11 years, 26% in ages 12-17 years, and 20% in ages 18-25 years.” 

4. How do you plan to address the issue of multiple comparisons and adjust for multiplicity?

The following text was added as the fourth paragraph of the statistical methods section to describe the plan for multiple comparisons and adjustments for multiplicity. 

“The study aims cover a wide range of scientific questions, but not all analyses will involve hypothesis testing. For instance, defining PASC does not require hypothesis testing, but evaluating whether the definition of or rates of PASC differ between age groups does. When multiple comparisons are made across age groups or other defined strata, or when different exposures and outcomes are assessed within the same subgroups, multiplicity adjustments for testing of non-exploratory hypotheses will be performed using the Hochberg procedure, in which tests of significance are performed in order of decreasing p-value with increasingly stringent thresholds. This approach has been found to sacrifice less power in observational studies with correlated outcomes compared to the Bonferroni and other standard approaches for addressing multiplicity.”

5. The determination of PASC is not clearly written. How will you validate the definition of PASC? What if the PASC cannot be differentiated, especially the sample size may be small in each age strata?

Thank you for this question. Since the initial submission of this study design paper, substantial work has been conducted within RECOVER to begin defining PASC in the adult cohort. This work will inform the working definition of PASC in children and young adults. 

The following text was added to the second paragraph of the statistical methods section to explain this:

“A preliminary working definition of PASC will be informed by using variable selection methods to identify which symptoms best differentiate children and young adults with and without an infection, following the methodology previously applied to develop a working definition of PASC in the RECOVER adult cohort. Data from the tier 1 visit will primarily be used.”

The following text was added to the second paragraph of the statistical methods section to explain this:

“The estimated associations obtained from regression models will be used to define a PASC score, with a cutoff for PASC defined based on clinical expertise while ensuring that the rate of those with no history of infection who are diagnosed as having PASC is reasonably low. This preliminary symptoms-based working definition is intended for research purposes and not a clinical diagnosis. It will be modified and augmented by clinical and subclinical findings as they become available. The working definition of PASC that is developed will also be validated in RECOVER participants who have linked EHR data against other definitions derived from EHR-based cohorts. While the working definition of PASC will be initially developed within each age strata, depending on the overlap of key symptoms that are identified that define PASC, some age groups may be aggregated in the interest of developing a more unified definition of PASC. To identify PASC phenotypes among children and young adults who are classified as having PASC defined by symptom patterns, we will use unsupervised learning methods to discover symptom clusters within each age strata (e.g., agglomerative hierarchical clustering [66] and consensus clustering [67]) to define PASC sub-phenotypes.” 

6. Statistical Analysis: Which statistical models do you plan to use to address the research questions? Logistic regression? Multiple regression? Will you use models for repeated measurements or survival models for time-to-event data?

The following text was added to the third paragraph of the statistical methods section to provide more details about our statistical analysis plan. 

“Logistic and poisson regression will be used to evaluate the association between pre-infection factors and PASC as a binary outcome, and multinomial regression will be used when PASC sub-type is used as a categorical outcome. Among participants in Tier 2 who develop PASC, we will use time-to-event analyses (i.e., Cox proportional hazards regression) to identify factors that influence time to recovery from PASC.”

Reviewer #3: 

Overall, this article provides a comprehensive overview of the RECOVER-Pediatrics study, which aims to characterize the clinical course, underlying mechanisms, and long-term health effects of post-acute sequelae of SARS-CoV-2 (PASC) in children and young adults. The article effectively highlights the unique challenges in understanding PASC symptoms in children and emphasizes the need for large-scale studies to define and recognize PASC in this population. The study design and methodology are well-described, and the article provides a clear outline of the study aims and objectives.

Minor comments:

7. The article would benefit from a clearer and more concise introduction. The current introduction provides a lot of statistics and information upfront, which can be overwhelming for the reader.

Consider restructuring the introduction to provide a brief overview of PASC in children and its impact before delving into the prevalence and incidence statistics.

We appreciate this feedback. We have added a brief overview of PASC in children at the beginning of the introduction. 

The following text was added at the beginning of the introduction:

“Long COVID, or the post-acute sequelae of SARS-CoV-2 (PASC), has been defined as symptoms, signs and conditions that continue or develop after a SARS-CoV-2 infection. These symptoms can affect people for weeks, months or even years after getting coronavirus disease 2019 (COVID-19). Symptoms can develop shortly after the initial recovery from an acute COVID-19 episode or persist from the initial illness. Symptoms may also emerge later or fluctuate or relapse over time. These symptoms can have debilitating effects on the daily health and quality of life of those affected.” 

8. Some of the sentences in the article are long and complex, which can make it difficult to follow the main points. Try breaking down these sentences into smaller, more digestible chunks to improve readability. For example, in the sentence "Given that young children might not be able to articulate symptoms, studies must rely on caregiver interpretation," breaking it into two sentences would make it easier to follow.

We have broken many of the longer sentences into multiple sentences in order to enhance the readability of the paper. 

For example, we divided the sentence noted above into two sentences. 

“For example, young children might not be able to articulate their symptoms. This has required studies to rely on caregiver interpretation of their young child’s symptoms.”

9. It would be helpful to provide more information about the selection process and criteria for the 10 hubs managing the study sites. 

The following information about the selection process was added to the end of the section called Study organizational structure and management. 

“Awardees were selected through a process that included independent peer review in response to OTA-21-015B.”

10. Additionally, mentioning the geographical distribution of the sites or regions covered would provide context and a better understanding of the study's reach.

The RECOVER-Pediatric study has wide geographic reach with sites located within 39 states, as well as Washington DC and Puerto Rico. The pediatric study also includes two large practice-based networks that are recruiting nationally in locations that may not have a larger site nearby. Supplemental Table 1 includes a list of all RECOVER-Pediatric sites and their locations. 

The following sentence was also added to the manuscript in the study organizational structure and management section. 

“RECOVER-Pediatrics includes 10 hubs that manage ~100 sites (Supplemental Table 1), located in more than 39 states, Washington DC and Puerto Rico.”

Reviewer #4: 

In this manuscript, Gross and colleagues present the protocol of the RECOVER-Pediatrics study, which aims to comprehensively characterize the clinical course, underlying mechanisms, and long-term effects of post-acute sequelae of SARS-CoV-2 from birth through 25 years old. Overall, the manuscript effectively introduces a well-designed protocol. However, I have a few suggestions for improvement:

11. The authors should consider adding a specific definition of post-acute sequelae of SARS-CoV-2 (PASC) in the first paragraph of the introduction. Providing a concise and clear definition would enhance the readers' understanding of the study's focus and objectives.

Thank you for this comment. This aligns with reviewer 3’s request in comment 7. 

The following concise and clear definition of PASC was added at the beginning of the introduction:

“Long COVID, or the post-acute sequelae of SARS-CoV-2 (PASC), has been defined as symptoms, signs and conditions that continue or develop after a SARS-CoV-2 infection. These symptoms can affect people for weeks, months or even years after getting coronavirus disease 2019 (COVID-19). Symptoms can develop shortly after the initial recovery from an acute COVID-19 episode or persist from the initial illness. Symptoms may also emerge later or fluctuate or relapse over time. These symptoms can have debilitating effects on the daily health and quality of life of those affected.” 

12. It would be beneficial if the authors could elaborate on how they plan to handle missing data. Addressing this aspect in the manuscript would enhance the transparency of the study's methodology.

Thank you for this comment. We added more detail about the study plan for handling missing data. 

The following text was added as the fifth paragraph of the statistical methods section:

“Potential sources of missing data include item nonresponse and attrition. Multiple imputation by chained equations will be the primary approach used to handle item nonresponse. Sensitivity analyses will include adjustments for potentially missing not at random data (i.e., informative missingness) using pattern mixture models, which permit the distribution of missing variables to differ between observed and unobserved values. Attrition, or missed visits, in the longitudinal phase of the study (Tier 2) will be addressed depending on the affected analysis. For time-to-event modeling, attrition induces censoring, which may not be independent if participants drop out of the study in a systematic fashion (i.e., participants with worse symptom trajectories may be less likely to continue to participate in the study). Inverse probability of censoring weights will be used to address dependent censoring in this context. For analyses with repeated measures, multiple imputation alongside likelihood-based methods, which are robust to the missing at random assumption, will be used to handle missing data, with pattern mixture models used to perform sensitivity analyses for informatively missing data as appropriate.”

13. In order to provide a more comprehensive understanding of the statistical analysis plan, the authors should consider including additional details, such as the software that will be used for the analysis and the specific statistical tests

---

## [Decision Letter · Decision Letter 1]

22 Nov 2023

Researching COVID to enhance recovery (RECOVER) pediatric study protocol: Rationale, objectives and design

PONE-D-23-10495R1

Dear Dr. Gross,

We’re pleased to inform you that your manuscript has been judged scientifically suitable for publication and will be formally accepted for publication once it meets all outstanding technical requirements.

Kind regards,

Seyed Aria Nejadghaderi

Academic Editor

PLOS ONE

Additional Editor Comments (optional):

Reviewers' comments:

Reviewer's Responses to Questions

**Comments to the Author**

1. Does the manuscript provide a valid rationale for the proposed study, with clearly identified and justified research questions?

Reviewer #2: Yes

Reviewer #4: Yes

2. Is the protocol technically sound and planned in a manner that will lead to a meaningful outcome and allow testing the stated hypotheses?

Reviewer #2: Yes

Reviewer #4: Yes

3. Is the methodology feasible and described in sufficient detail to allow the work to be replicable?

Reviewer #2: Yes

Reviewer #4: Yes

4. Have the authors described where all data underlying the findings will be made available when the study is complete?

Reviewer #2: Yes

Reviewer #4: Yes

5. Is the manuscript presented in an intelligible fashion and written in standard English?

Reviewer #2: Yes

Reviewer #4: Yes

6. Review Comments to the Author

You may also provide optional suggestions and comments to authors that they might find helpful in planning their study.

Reviewer #2: All questions I raised were addressed. I have no more comments. The protocol is good for publication now.

Reviewer #4: The authors have addressed all my comments. I have no further comments. I thank them for their detailed revision.

7. PLOS authors have the option to publish the peer review history of their article (what does this mean?). If published, this will include your full peer review and any attached files.

Reviewer #2: No

Reviewer #4: No
